# Online Gradient Boosting Decision Tree: In-Place Updates for Adding/Deleting Data

## Abstract

Gradient Boosting Decision Tree (GBDT) is one of the most popular machine learning models in various applications. But in the traditional settings, all data should be simultaneously accessed in the training procedure: it does not allow to add or delete any data instances after training. In this paper, we propose a novel online learning framework for GBDT supporting both incremental and decremental learning. To the best of our knowledge, this is the first work that considers an in-place unified incremental and decremental learning on GBDT. To reduce the learning cost, we present a collection of optimizations for our framework, so that it can add or delete a small fraction of data on the fly. We theoretically show the relationship between the hyper-parameters of the proposed optimizations, which enables trading off accuracy and cost on incremental and decremental learning. The backdoor attack results show that our framework can successfully inject and remove backdoor in a well-trained model using incremental and decremental learning, and the empirical results on public datasets confirm the effectiveness and efficiency of our proposed online learning framework and optimizations.

## 1 Introduction

Gradient Boosting Decision Tree (GBDT) has demonstrated outstanding performance across a wide range of applications Sudakov et al. (2019); Biau et al. (2019); Rao et al. (2019); Liu & Yu (2007). It outperforms deep learning models on many datasets in accuracy and provides interpretability for the trained models. However, in traditional setting, all data is simultaneously accessed in training procedure, which makes its application scenarios limited.

**Online Learning.** Online learning is a machine learning approach where data is sequentially available and used to update the predictor for the latest data Bertsekas (2015); Parisi et al. (2019); Hazan (2016); Oza (2005). Generally, online learning is expected to possess the capabilities of both incremental learning (adding training data) and decremental learning (removing a subset of training data). This allows the model to dynamically adapt to the latest data while removing outdated data.

**Incremental Learning.** There are some challenges for incremental learning in GBDT due to its natural properties Friedman et al. (2000). Traditional GBDT trains over an entire dataset, and each node is trained on the data reaching it to achieve the best split for optimal accuracy. Adding unseen data may affect node splitting results, potentially leading to catastrophic performance changes.

Moreover, training gradient boosting models involves creating trees for each iteration, with tree fitting based on the residual of previous iterations. More iterations create more trees, increasing model sizes and hurting inference throughput. This also prohibits tasks like fine-tuning or transfer learning without substantially increasing model sizes.

Recent studies have explored incremental learning on classic machine learning, such as SVM, random forest, and gradient boosting. Shilton et al. (2005); Laskov et al. (2006); Fine & Scheinberg (2001) proposed methods to maintain SVM optimality after adding a few training vectors. Wang et al. (2009) presented an incremental random forest for online learning with small streaming data. Beygelzimer et al. (2015a) extended gradient boosting theory for regression to online learning. Zhang et al. (2019) proposed iGBDT for incremental learning by "lazily" updating, but it may require retraining many trees when the new data size is large. It is important to note that prior studies on online gradient boosting Beygelzimer et al. (2015a); Chen et al. (2012); Beygelzimer et al. (2015b) and incremental gradient boosting Zhang et al. (2019); Hu et al. (2017) do not support decremental learning.

---

**Algorithm 1** Robust LogitBoost Algorithm.

1: $F_{i,k} = 0, p_{i,k} = \frac{1}{K}, k = 0$ to $K-1, i = 1$ to $N$
2: **for** $m = 0$ to $M-1$ **do**
3:   **for** $k = 0$ to $K-1$ **do**
4:     $\hat{D}_{tr} = \{r_{i,k} - p_{i,k}, \ \mathbf{x}_i\}_{i=1}^N$
5:     $w_{i,k} = p_{i,k}(1 - p_{i,k})$
6:     $\{R_{j,k,m}\}_{j=1}^J = J$-terminal node regression tree from $\hat{D}_{tr}$, with weights $w_{i,k}$, using the tree split gain formula Eq. equation 5.
7:     $\beta_{j,k,m} = \frac{K-1}{K} \frac{\sum_{\mathbf{x}_i \in R_{j,k,m}} r_{i,k} - p_{i,k}}{\sum_{\mathbf{x}_i \in R_{j,k,m}} (1 - p_{i,k}) p_{i,k}}$
8:     $f_{i,k} = \sum_{j=1}^J \beta_{j,k,m} \mathbb{1}_{\mathbf{x}_i \in R_{j,k,m}}, \quad F_{i,k} = F_{i,k} + \nu f_{i,k}$
9:   **end for**
10:   $p_{i,k} = \exp(F_{i,k}) / \sum_{s=1}^K \exp(F_{i,s})$
11: **end for**

---

**Algorithm 2** Online Learning in Gradient Boosting

1: $D' = D_{in}$ **if** incremental learning **else** $D_{de}$
2: **for** $m = 0$ to $M-1$ **do**
3:   **for** $k = 0$ to $K-1$ **do**
4:     $\hat{D}' = \{r_{i,k} - p_{i,k}, \ \mathbf{x}_i\}_{i=1}^{|D'|}$
5:     Compute $w_{i,k} = p_{i,k}(1 - p_{i,k})$ for $\hat{D}'$ using $F_{i,k}$
6:     Compute $r_{i,k}$ for $\hat{D}'$ using $F_{i,k}$
7:     **if** incremental learning **then**
8:       $\left\{\hat{R}_{j,k,m}\right\}_{j=1}^J = \mathrm{incr}(\{R_{j,k,m}\}_{j=1}^J, \hat{D}', w_{i,k}, r_{i,k})$
9:     **else**
10:       $\left\{\hat{R}_{j,k,m}\right\}_{j=1}^J = \mathrm{decr}(\{R_{j,k,m}\}_{j=1}^J, \hat{D}', w_{i,k}, r_{i,k})$
11:     **end if**
12:     Update $F_{i,k}$ with $\left\{\hat{R}_{j,k,m}\right\}_{j=1}^J$
13:   **end for**
14: **end for**

---

**Decremental Learning.** Decremental learning is more complex and less studied than incremental learning. Cauwenberghs & Poggio (2000) presented an online recursive algorithm for training SVM with an efficient decremental learning method. Chen et al. (2019) proposed online incremental and decremental learning algorithms based on variable SVM, leveraging pre-calculated results. Brophy & Lowd (2021) and Brophy & Lowd (2020) provided methods for data addition and removal in random forests. Schelter et al. (2021) proposed robust tree node split criteria and alternative splits for low-latency unlearning. Many works have also studied decremental learning in deep neural networks (DNN). Bourtoule et al. (2021) introduced a framework that accelerates decremental learning by constraining individual data points' impact during training.

While online learning has emerged as a popular topic recently, it has been barely investigated on GBDT. Wu et al. (2023); Lin et al. (2023) are among the latest studies in decremental learning for GBDT. Wu et al. (2023) presented DeltaBoost, a GBDT-like model enabling data deletion. DeltaBoost uses bagging to divide the training dataset into disjoint sub-datasets, training each iteration's tree on a different sub-dataset, reducing tree dependency. However, this simplification may impact model performance. Lin et al. (2023) proposed an unlearning framework in GBDT without simplification, unlearning specific data using recorded auxiliary information from training. It optimizes to reduce unlearning time, making it faster than retraining from scratch, but introduces many hyper-parameters and performs poorly on extremely large datasets.

In this paper, we propose a novel incremental and decremental learning framework for GBDT. To the best of our knowledge, this is the first work that considers in-place online incremental and decremental learning at the same time on GBDT. The incremental and decremental learning in this work applies a unified notion, which enables convenient implementation.

**Challenges.** We identify three major challenges of in-place online learning for GBDT: (1) Unlike batch training of deep neural networks (DNN), more iterations in GBDT create more trees and parameters, leading to unbounded memory and computation costs in online learning. In-place learning on originally constructed trees is necessary for practicality. (2) Gradient-based methods in DNN add and subtract gradients for incremental and decremental learning, but GBDT is not differentiable. (3) GBDT depends on the residual of the previous tree, unlike independent iterations in random forests. Changing one tree requires modifying all subsequent trees, complicating incremental and decremental learning.

**Contributions.** (1) We introduce a novel in-place online learning framework for gradient boosting models supporting incremental and decremental learning, extensible to fine-tuning and transfer learning. (2) We present optimizations to reduce the cost of incremental and decremental learning, making adding or deleting a small data fraction substantially faster than retraining. (3) We theoretically show the relationship among optimization hyper-parameters, enabling trade-offs between accuracy and cost. (4) We experimentally evaluate our framework on public datasets, confirming its effectiveness and efficiency. (5) We release an open-source implementation of our framework[1].

## 2 ONLINE GBDT FRAMEWORK

### 2.1 GBDT PRELIMINARY

Gradient Boosting Decision Tree (GBDT) is an powerful ensemble technique that combines multiple decision tree to produce an accurate predictive model (Friedman et al., 2000; Friedman, 2001).

---

[1] https://anonymous.4open.science/r/In-PlaceOnlineGBDT

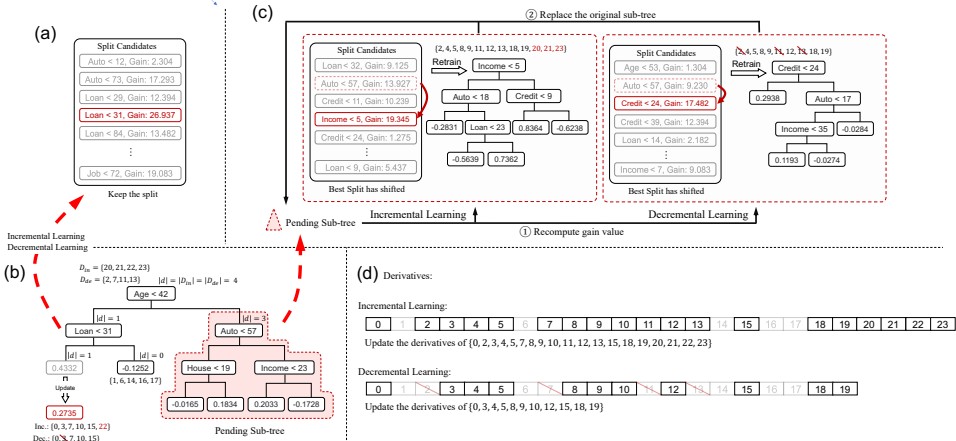

Figure 1: An example for the incremental learning and decremental learning procedure in the proposed framework. (a) For the node of `Loan < 31`, the current split is still the best after online learning. Thus, the split does not need to change. (b) An already well-trained tree in $D_{tr}$. (c) For the node of `Auto < 57`, the best split has shifted after online learning. (d) Incremental update for derivatives – only update the derivatives for those data reaching the changed terminal nodes.

Given a dataset $D_{tr} = \{y_i, \mathbf{x}_i\}_{i=1}^N$, where $N$ is the size of training dataset, and $\mathbf{x}_i$ indicates the $i^{th}$ data vector and $y_i \in \{0, 1, ..., K-1\}$ denotes the label for the $i^{th}$ data point. For a GBDT model with $M$ iteration, the probability $p_{i,k}$ for $i^{th}$ data and class $k$ is:

$$p_{i,k} = \mathbf{Pr}\left(y_i = k | \mathbf{x}_i\right) = \frac{e^{F_{i,k}(\mathbf{x_i})}}{\sum_{s=1}^K e^{F_{i,s}(\mathbf{x_i})}}, \qquad i = 1, 2, ..., N \tag{1}$$

where $F$ is a combination of M terms:

$$F^{(M)}(\mathbf{x}) = \sum_{m=0}^{M-1} \rho_m h(\mathbf{x}; \mathbf{a}_m) \tag{2}$$

where $h(\mathbf{x}; \mathbf{a}_m)$ is a regression tree, and $\rho_m$ and $\mathbf{a}_m$ denote the tree parameters that learned by minimizing the *negative log-likelihood*:

$$L = \sum_{i=1}^N L_i, \qquad L_i = - \sum_{k=0}^{K-1} r_{i,k} \log p_{i,k} \tag{3}$$

where $r_{i,k} = \begin{cases} 1, & \text{if } y_i = k \\ 0, & \text{otherwise} \end{cases}$. The training procedures require calculating the derivatives of loss function $L$ with respect to $F_{i,k}$:

$$g_{i,k} = \frac{\partial L_i}{\partial F_{i,k}} = -\left(r_{i,k} - p_{i,k}\right), \quad h_{i,k} = \frac{\partial^2 L_i}{\partial F_{i,k}^2} = p_{i,k}\left(1 - p_{i,k}\right). \tag{4}$$

In GBDT training, to solve numerical instability problem (Friedman et al., 2000; Friedman, 2001; Friedman et al., 2008), we apply **Robust LogitBoost** algorithm (Li, 2010) as shown in Algorithm 1, which has three parameters, the number of terminal nodes $J$, the shrinkage $\nu$ and the number of boosting iterations $M$. To find the optimal split for a decision tree node, we first sort the $N$ data by the feature values being considered for splitting. We then iterate through each potential split index $s$, where $1 \le s < N$, to find the best split that minimizes the weighted squared error (SE) between the predicted and true labels. Specifically, we aim to find an split $s$ to maximize the gain function:

$$Gain(s) = \frac{\left(\sum_{i=1}^s g_{i,k}\right)^2}{\sum_{i=1}^s h_{i,k}} + \frac{\left(\sum_{i=s+1}^N g_{i,k}\right)^2}{\sum_{i=s+1}^N h_{i,k}} - \frac{\left(\sum_{i=1}^N g_{i,k}\right)^2}{\sum_{i=1}^N h_{i,k}}. \tag{5}$$

## 2.2 PROBLEM SETTING

For classic GBDT, all training data must be loaded during training, and adding/deleting instances is not allowed afterwards. This work proposes an online GBDT framework that enables in-place addition/deletion of specific data instances to/from a well-trained model through incremental and decremental learning.

**Problem Statement.** Given a trained gradient boosting model $T(\theta)$ on training dataset $D_{tr}$, where $\theta$ indicates the parameters of model $T$, an incremental learning dataset $D_{in}$, and/or a decremental

learning dataset $D_{de}$ ($D_{de} \subseteq D_{tr}$), our goal is to find a tree model $T(\theta')$ that fits dataset $D_{tr} \cup D_{in} \setminus D_{de}$, where $|\theta| = |\theta'|$ (the parameter size and the number of trees stay unchanged).

The most obvious way is to retrain the model from scratch on dataset $D_{tr} \cup D_{in} \setminus D_{de}$. However, retraining is time-consuming and resource-intensive. Especially for online learning applications, rapid retraining is not practical. The key question of this problem is: ***Can we obtain the model $T(\theta')$ based on the learned knowledge of the original model $T(\theta)$ without retraining the entire model?***

The proposed framework aims to find a tree model $T(\theta')$ as close to the model retraining from scratch as possible based on the learned knowledge of the model $T(\theta)$. In addition, this online learning algorithm is in a "warm-start" manner, because it learns a new dataset $D_{in}$ or removes a learned sub-dataset $D_{de} \subseteq D_{tr}$ on a model that is already well-trained on training dataset $D_{tr}$.

Let $\mathcal{A}$ denotes the initial GBDT learning algorithm , then we have $\mathcal{A}(D_{tr}) \in \mathcal{H}$, where $\mathcal{H}$ is the hypothesis space. An online learning algorithm $\mathcal{L}$ for incremental learning or decremental learning can be used to learn dataset $D_{in}$ or remove dataset $D_{de} \subseteq D_{tr}$.

### 2.3 FRAMEWORK OVERVIEW

The goal of this work is to propose an online learning framework for GBDT that supports incremental learning and decremental learning for any collection of data.

**Online Learning in GBDT.** The Algorithm 2 shows the procedure of online learning in GBDT. At first, the GBDT model is a well-trained model on the training dataset $D_{tr}$. Recall that the GBDT model is frozen and can not be changed after training—no training data modification. In this proposed framework, the user can do (1) incremental learning: update a new dataset $D_{in}$ to the model, and (2) decremental learning: remove a learned dataset $D_{de} \subseteq D_{tr}$ and its effect on the model.

As shown in Algorithm 2, it is similar to the learning process, but it only needs to compute $r_{i,k}$ and $p_{i,k}(1 - p_{i,k})$ for online dataset $D'$ without touching the training dataset $D_{tr}$. Then, it will call the function of incremental learning or decremental learning to obtain $\left\{ \hat{R}_{j,k,m} \right\}_{j=1}^{J}$. Finally, we update $F_{i,k}$ with new $\left\{ \hat{R}_{j,k,m} \right\}_{j=1}^{J}$. Here we use the same notion to design the function of incremental learning and decremental learning – decremental learning is the inverse process of incremental learning for dataset $D'$. Therefore, we describe them in the Algorithm 3 at the same time.

**Incremental & Decremental Learning on One Tree.** Algorithm 3 describes the detailed process for incremental and decremental learning, which are almost the same as decremental learning is the inverse of incremental learning for dataset $D'$. The main difference is at Line 3. First, we traverse all non-terminal nodes with ascending depths. For each node, let $s$ denote the current split. We recompute the new best gain value with $r_{i,k}$ and $p_{i,k}(1 - p_{i,k})$ after adding $D'$ for incremental learning or removing $D'$ for decremental learning. If the current split $s$ matches the new best split $s'$ (after adding/removing $D'$), we keep the current split (Figure 1(a)). Otherwise, if the current best split has changed ($s \neq s'$), as shown in Figure 1(c),

---

**Algorithm 3** Incr./Decr. Learning on One Tree

1. **for** non-terminal *node* **in** $\{R_{j,k,m}\}_{j=1}^{J}$ with ascending depths **do**
2. $\quad \hat{D}' = \{r_{i,k} - p_{i,k}, \ \mathbf{x}_i\}_{i=1}^{|D'|}$
3. $\quad s$ = current split of *node*
4. $\quad s'$ = compute best gain with Eq. equation 5 with $r_{i,k}$ and $w_{i,k}$ after adding/removing $\hat{D}'$
5. $\quad$ **if** $s' \neq s$ **then**
6. $\qquad$ Retrain the subtree rooted at *node*.
7. $\quad$ **end if**
8. **end for**
9. Update prediction value $\beta_{j,k,m}$ for all terminal nodes

---

we retrain the sub-tree rooted on this node and replace it with the new sub-tree. After testing all nodes, node splits remain on the best split. Finally, we recompute the prediction value on all terminal nodes. Appendix C provides a detailed explanation of Figure 1.

## 3 OPTIMIZING LEARNING TIME

In this section, we introduce optimizations for our online learning framework to reduce computation overhead and costs. The key step is deciding whether a node should be kept or replaced: *Can we design an algorithm to quickly test whether the node should be retained or retrained without touching the training data?* Our most important optimization is to avoid touching the full training dataset. We apply incremental update and split candidates sampling concepts from Lin et al. (2023), extend them to support online learning, and provide evidence of the relationship between hyper-parameters

of different optimizations, enabling trade-offs between accuracy and cost. Additionally, we design optimizations specific to online learning: 1) adaptive lazy update for residuals and hessians to substantially decrease online learning time; 2) adaptive split robustness tolerance to significantly reduce the number of retrained nodes.

## 3.1 Update without Touching Training Data

To reduce computation overhead and online learning time, we target to avoid touching the original training dataset $D$, and only focus on the online learning dataset $D'$. Following the study Lin et al. (2023), we extend the optimization of updating statistical information to the scenarios of online learning: (1) Maintain Best Split; (2) Recomputing Prediction Value; (3) Incremental Update for Derivatives, and the computation cost is reduced from $O(D \pm D')$ to $O(D')$ by these optimizations. The implementation of these optimizations are included in Appendix E.

## 3.2 Adaptive Lazy Update for Derivatives

Although incremental update can substantially reduce online learning time, we can take it a step further: if no retraining occurs, the changes to the derivatives will be very small. *How can we effectively utilize the parameters already learned to reduce online learning time?*

Gradient Accumulation Li et al. (2014); Goyal et al. (2017); Ruder (2016) is widely used in DNN training. After computing the loss and gradients for each mini-batch, the system accumulates these gradients over multiple batches instead of updating the model parameters immediately. Inspired by Gradient Accumulation techniques, we introduce an adaptive lazy update for our online learning framework. Unlike Lin *et al.* Lin et al. (2023), which perform updates after a fixed number of batches, we update the derivatives only when retraining occurs. This approach uses more outdated derivatives for gain computation but significantly reduces the cost of derivative updates.

## 3.3 Split Candidates Sampling

From the above optimizations, if retraining is not required, we can keep the current best split. In this case, we only need to iterate over the online learning dataset $D'$ and update the prediction values to accomplish online learning, whether it involves adding or removing data. However, if the sub-tree rooted in this node requires retraining, it is necessary to train the new sub-tree on the data from the dataset $D_{tr} \pm D'$ that reaches this node. It is clear that retraining incurs more resource consumption and takes a longer execution time. In the worst case, if retraining is required in the root node, it has to retrain the entire new tree on full dataset $D_{tr} \pm D'$.

To reduce the time and resource consumption of online learning, a straightforward approach is to minimize the frequency of retraining. Therefore, we introduce split candidate sampling to reduce frequent retraining by limiting the number of splits, and it is beneficial for both training and online learning. All features are discretized into integers in $\{0, 1, 2, \cdots, B-1\}$, as shown in Appendix A. The original training procedure enumerates all $B$ potential splits, and then obtains the best split with the greatest gain value. In split candidates sampling, we randomly select $\lceil \alpha B \rceil$ potential splits as candidates and only perform gain computing on these candidates. As $\alpha$ decreases, the number of split candidates decreases, resulting in larger distances between split candidates. Consequently, the best split is less likely to change frequently.

**Definition 1** (Distance Robust) *Let $s$ be the best split, and $\frac{|D'|}{|D_{tr}|} = \lambda$. $N_\Delta$ is the distance between $s$ and its nearest split $t$ with the same feature, $N_\Delta = ||t - s||$. $s$ is distance robust if*

$$N_\Delta > \frac{\lambda Gain(s)}{\frac{1}{N_{ls}} \frac{\left(\sum_{\mathbf{x}_i \in l_s} g_{i,k}\right)^2}{\sum_{\mathbf{x}_i \in l_s} h_{i,k}} + \frac{1}{N_{rs}} \frac{\left(\sum_{\mathbf{x}_i \in r_s} g_{i,k}\right)^2}{\sum_{\mathbf{x}_i \in r_s} h_{i,k}}} \tag{6}$$

where $l$ represents the left child of split $s$, and it contains the samples belonging to this node, while $r$ represent the right child, $N_{ls}$ denotes $|l_s|$, and $N_{rs}$ denotes $|r_s|$. In this definition, $\mathbb{E}(N_\Delta) = 1/\alpha$, where $\alpha$ denotes the split sampling rate, we can observe that a smaller sampling rate will result in a more robust split, so we can reduce the number of retrain operations by reducing the sampling rate. Similarly, incremental learning can get the same result.

**Definition 2** (Robustness Split) *For a best split $s$ and an arbitrary split $t, t \neq s$, and online learning data rate $\frac{|D'|}{|D_{tr}|} = \lambda$, the best split $s$ is robust split if*

$$Gain(s) > \frac{1}{1 - \lambda} Gain(t) \tag{7}$$

Robustness split shows that, as $\lambda = \frac{|D'|}{|D_{tr}|}$ decreases, the splits are more robust, leading to a reduction in the frequency of retraining. In conclusion, decreasing either $\alpha$ or $\lambda$ makes the split more robust, reducing the change occurrence in the best split, and it can significantly reduce the online learning time. We provide the proof of *Distance Robust* and *Robustness Split* in Appendix D.

### 3.4 ADAPTIVE SPLIT ROBUSTNESS TOLERANCE

Recall the retraining condition for a node that we mentioned previously: we retrain the subtree rooted at a node if the best split changes. Although the best split may have changed to another one, the gain value might only be slightly different from the original best split. We illustrate the observation of the distance of best split changes (the changes in the ranking of the best split) in Figure 2. The top row illustrates the distance of best split changes observed in the Adult and Covtype datasets for incremental learning, while the bottom row depicts same in Letter and SUSY datasets for decremental learning. Similar patterns are observed across various other datasets. For adding or deleting a single data point, the best split does not change in most cases. As the $|D'|$ increases to $0.1\%$, $0.5\%$, and $1\%$, the best split in most cases switch to the second best. If we only apply the optimal split, it will lead to frequent retraining during online learning.

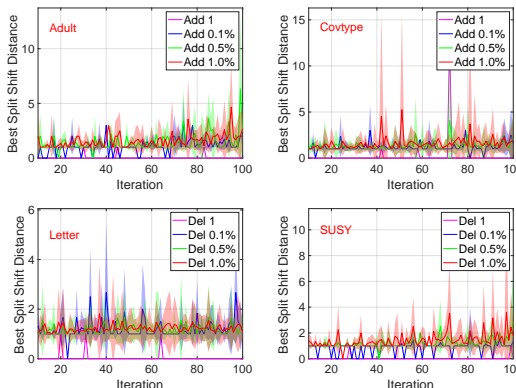

Figure 2: Observation of distance of best split changes. The lines represents the average changes of best split distance, and the shaded region is the standard error.

The distance of the best split changes is usually small. Tolerating its variation within a certain range and continuing to use the original split significantly accelerates online learning. We propose adaptive split robustness tolerance: for a node with $\lceil \alpha B \rceil$ potential splits, if the current split is among the top $\lceil \sigma \alpha B \rceil$, we continue using it, where $\sigma$ $(0 \leq \sigma \leq 1)$ is the robustness tolerance. $\sigma = 0$ selects only the best split, while $\sigma = 1$ avoids retraining. Higher $\sigma$ indicates greater tolerance, making the split more robust and less likely to retrain.

## 4 EXPERIMENTAL EVALUATION

In this section, we report empirical evaluation in different aspects. We compare our 1) incremental learning with OnlineGradientBoost (OnlineGB)[2] Leistner et al. (2009) and iGBDT Zhang et al. (2019); 2) decremental learning with DeltaBoost[3] Wu et al. (2023) and MUinGBDT[4] Lin et al. (2023); 3) training cost with popular GBDT libraries XGBoost Chen & Guestrin (2016), LightGBM Ke et al. (2017), CatBoost Dorogush et al. (2018) and ThunderGBM Wen et al. (2020).

**Implementation Details.** The details of environments and settings are included in Appendix B. We employ one thread for all experiments to have a fair comparison, and run ThunderGBM on a NVIDIA A100 40GB GPU, since it does not support only CPU[5]. Unless explicitly stated otherwise, our default parameter settings are as follows: $\nu = 1$, $M = 100$, $J = 20$, $B = 1024$, $|D'| = 0.1\% \times |D_{tr}|$, $\alpha = 0.1$, and $\sigma = 0.1$.

Table 1: Dataset specifications.

| Dataset | # Train | # Test | # Dim | # Class |
|---|---|---|---|---|
| Adult | 36,139 | 9,034 | 87 | 2 |
| CreditInfo | 105,000 | 45,000 | 10 | 2 |
| SUSY | 2,500,000 | 2,500,000 | 18 | 2 |
| HIGGS | 5,500,000 | 5,500,000 | 28 | 2 |
| Optdigits | 3,822 | 1,796 | 64 | 10 |
| Pendigits | 7,493 | 3,497 | 16 | 10 |
| Letter | 15,000 | 5,000 | 16 | 26 |
| Covtype | 290,506 | 290,506 | 54 | 7 |
| Abalone | 2,785 | 1,392 | 8 | Reg. |
| WineQuality | 4,332 | 2,165 | 12 | Reg. |

**Datasets.** We utilize 10 public datasets in the experiments. The specifications of these datasets are presented in Table 1. The smallest dataset, Optdigits, consists of 3,822 training instances, while the largest dataset, HIGGS, contains a total of 11 million instances. The number of dimensions or features varies between 8 and 87 across the datasets.

### 4.1 TRAINING TIME AND MEMORY OVERHEAD

Since the proposed online learning framework stores certain statistical information during training, this may impact both the training time and memory usage. Table 2 presents a detailed report of the total training time and memory overhead.

---

[2] https://github.com/charliermarsh/online_boosting [3] https://github.com/Xtra-Computing/DeltaBoost/
[4] https://github.com/huawei-lin/GBDT_unlearning
[5] https://github.com/Xtra-Computing/thundergbm/blob/master/docs/faq.md

Table 3: Total incremental or decremental learning time (seconds). For the methods supporting incremental or decremental learning (OnlineGB, iGBDT, DeltaBoost, MU in GBDT), $speedup = \frac{incr./decr.\ learning\ time}{our\ online\ learning\ time}$, otherwise, $speedup = \frac{training\ time}{our\ online\ learning\ time}$.

| Dataset | $|D'|$ | Inc. Total Time: OnlineGB | iGBDT | Ours | Inc. Speedup: OnlineGB | iGBDT | XGBoost | LightGBM | CatBoost | ThunderGBM (GPU) | Dec. Total Time: DeltaBoost | MU in GBDT | Ours | Dec. Speedup: DeltaBoost | MU in GBDT | XGBoost | LightGBM | CatBoost | ThunderGBM (GPU) |
|---|---|---|---|---|---|---|---|---|---|---|---|---|---|---|---|---|---|---|---|
| Adult | 1 | 0.265 | 0.595 | **0.035** | 7.6x | 17x | 270.5x | 14.7x | 43.8x | 16.1x | 0.923 | 0.217 | **0.034** | 27.1x | 6.4x | 278.4x | 15.2x | 45.1x | 16.6x |
| | 0.1% | 9.02 | 1.145 | **0.105** | 85.9x | 10.9x | 90.2x | 4.9x | 14.6x | 5.4x | 28.022 | 0.751 | **0.103** | 272.1x | 7.3x | 91.9x | 5x | 14.9x | 5.5x |
| | 0.5% | 44.65 | 1.296 | **0.212** | 210.6x | 6.1x | 44.7x | 2.4x | 7.2x | 2.7x | 34.461 | 1.059 | **0.222** | 155.2x | 4.8x | 42.6x | 2.3x | 6.9x | 2.3x |
| | 1% | 98 | 1.573 | **0.344** | 284.9x | 4.6x | 27.5x | 1.5x | 4.5x | 1.6x | 62.124 | 1.276 | **0.379** | 163.9x | 3.4x | 25x | 1.4x | 4x | 1.5x |
| CreditInfo | 1 | 29 | 0.475 | **0.114** | 254.4x | 4.2x | 116.8x | 16.1x | 30.2x | 5.1x | 89.097 | 0.113 | **0.055** | 1,619.9x | 2.1x | 242.1x | 33.4x | 62.7x | 10.6x |
| | 0.1% | 3,386.25 | 1.391 | **0.249** | 13,599.4x | 5.6x | 53.5x | 7.4x | 13.8x | 2.3x | 78.836 | 0.426 | **0.153** | 515.3x | 2.8x | 87x | 12x | 22.5x | 3.8x |
| | 0.5% | 28,875 | 1.428 | **0.321** | 89,953.3x | 4.4x | 41.5x | 5.7x | 10.7x | 1.8x | 80.559 | 0.824 | **0.251** | 321x | 3.3x | 53x | 7.3x | 13.7x | 2.3x |
| | 1% | 336,000 | 1.568 | **0.383** | 877,284.6x | 4.1x | 34.8x | 4.8x | 9x | 1.5x | 74.331 | 1.065 | **0.355** | 209.4x | 3x | 37.5x | 5.2x | 9.7x | 1.6x |
| SUSY | 1 | OOM | 12.037 | **1.678** | - | 7.2x | 974.3x | 58.2x | 64.9x | 3.6x | 309.19 | 1.707 | **1.303** | 237.3x | 1.3x | 1,254.7x | 74.9x | 83.6x | 4.6x |
| | 0.1% | OOM | 53.46 | **7.972** | - | 6.7x | 205.1x | 12.2x | 13.7x | 0.8x | 180.894 | 23.999 | **6.263** | 28.9x | 3.8x | 261x | 15.6x | 17.4x | 1x |
| | 0.5% | OOM | 55.38 | **13.39** | - | 4.1x | 122.1x | 7.3x | 8.1x | 0.4x | 197.86 | 53.962 | **15.438** | 12.8x | 3.5x | 105.9x | 6.3x | 7.1x | 0.4x |
| | 1% | OOM | 57.68 | **20.093** | - | 2.9x | 81.4x | 4.9x | 5.4x | 0.3x | 298.44 | 77.76 | **25.98** | 11.5x | 3x | 62.9x | 3.8x | 4.2x | 0.2x |
| HIGGS | 1 | OOM | 45.25 | **5.488** | - | 8.2x | 406.3x | 38.4x | 55.3x | 2.5x | OOM | 4.967 | **3.367** | - | 1.5x | 662.3x | 62.7x | 90.2x | 4.1x |
| | 0.1% | OOM | 132.46 | **26.558** | - | 5x | 84x | 7.9x | 11.4x | 0.5x | OOM | 55.265 | **18.926** | - | 2.9x | 117.8x | 11.1x | 16x | 0.7x |
| | 0.5% | OOM | 165.34 | **43.17** | - | 3.8x | 51.7x | 4.9x | 7x | 0.3x | OOM | 152.095 | **48.683** | - | 3.1x | 45.8x | 4.3x | 6.2x | 0.3x |
| | 1% | OOM | 171.16 | **65.579** | - | 2.6x | 34x | 3.2x | 4.6x | 0.2x | OOM | 251.224 | **80.776** | - | 3.1x | 27.6x | 2.6x | 3.8x | 0.2x |
| Optdigits | 1 | 0.032 | 0.174 | **0.011** | 2.9x | 15.8x | 68.4x | 9.6x | 16.1x | 26.9x | 0.687 | 0.015 | **0.01** | 68.7x | 1.5x | 75.2x | 10.6x | 17.7x | 29.6x |
| | 0.1% | 0.091 | 0.181 | **0.015** | 6.1x | 12.1x | 50.1x | 7.1x | 11.8x | 19.7x | 0.645 | 0.032 | **0.014** | 46.1x | 2.3x | 53.7x | 7.6x | 12.6x | 21.1x |
| | 0.5% | 0.559 | 0.191 | **0.029** | 19.3x | 6.6x | 25.9x | 3.7x | 6.1x | 10.2x | 0.563 | 0.067 | **0.029** | 19.4x | 2.3x | 25.9x | 3.7x | 6.1x | 10.2x |
| | 1% | 1.403 | 0.196 | **0.043** | 32.6x | 4.6x | 17.5x | 2.5x | 4.1x | 6.9x | 0.638 | 0.086 | **0.046** | 13.9x | 1.8x | 16.3x | 2.3x | 3.8x | 6.4x |
| Pendigits | 1 | **0.014** | 0.181 | **0.014** | 1x | 12.9x | 41x | 9.4x | 13.1x | 27.6x | 0.525 | **0.013** | 0.015 | 35x | 0.9x | 38.3x | 8.7x | 12.2x | 25.8x |
| | 0.1% | 0.082 | 0.224 | **0.026** | 3.2x | 8.6x | 22.1x | 5x | 7x | 14.9x | 0.465 | **0.022** | 0.025 | 18.6x | 0.9x | 23x | 5.2x | 7.3x | 15.5x |
| | 0.5% | 0.427 | 0.234 | **0.042** | 10.2x | 5.6x | 13.7x | 3.1x | 4.4x | 9.2x | 0.531 | 0.089 | **0.041** | 13x | 2.2x | 14x | 3.2x | 4.5x | 9.4x |
| | 1% | 0.82 | 0.235 | **0.053** | 15.5x | 4.4x | 10.8x | 2.3x | 3.5x | 7.3x | 0.768 | 0.129 | **0.057** | 13.5x | 2.3x | 10.1x | 2.3x | 3.2x | 6.8x |
| Letter | 1 | 0.033 | 0.102 | **0.016** | 2.1x | 6.4x | 73.2x | 12.7x | 14.5x | 22.9x | 0.863 | 0.017 | **0.014** | 61.6x | 1.2x | 83.6x | 14.5x | 16.6x | 26.1x |
| | 0.1% | 0.551 | 0.167 | **0.04** | 13.8x | 4.2x | 29.3x | 5.1x | 5.8x | 9.2x | 0.664 | **0.032** | 0.058 | 11.4x | 0.6x | 20.2x | 3.5x | 4x | 6.3x |
| | 0.5% | 2.768 | 0.187 | **0.067** | 41.3x | 2.8x | 17.5x | 3x | 3.8x | 5.5x | 0.676 | **0.066** | 0.103 | 6.6x | 0.6x | 11.4x | 2x | 2.3x | 3.6x |
| | 1% | 5.68 | 0.202 | **0.128** | 44.4x | 1.6x | 9.1x | 1.6x | 1.8x | 2.9x | 0.997 | **0.094** | 0.134 | 7.4x | 0.7x | 8.7x | 1.5x | 1.7x | 2.7x |
| Covtype | 1 | 0.09 | 1.321 | **0.29** | 0.3x | 4.6x | 220.4x | 15.8x | 21.2x | 5.1x | 28.519 | 0.562 | **0.161** | 177.1x | 3.5x | 397x | 28.5x | 38.1x | 9.2x |
| | 0.1% | 21.408 | 6.391 | **0.639** | 33.5x | 10x | 100x | 7.2x | 9.6x | 2.3x | 19.61 | 3.44 | **0.546** | 35.9x | 6.3x | 117.1x | 8.4x | 11.2x | 2.7x |
| | 0.5% | 105.688 | 7.765 | **1.095** | 96.5x | 7.1x | 58.4x | 4.2x | 5.6x | 1.3x | 20.035 | 5.519 | **1.187** | 16.9x | 4.6x | 53.8x | 3.9x | 5.2x | 1.2x |
| | 1% | 214.188 | 8.088 | **1.724** | 124.2x | 4.7x | 37.1x | 2.7x | 3.6x | 0.9x | 21.864 | 6.917 | **1.963** | 11.1x | 3.5x | 32.6x | 2.3x | 3.1x | 0.8x |
| Abalone | 1 | **0.013** | 0.331 | 0.027 | 0.5x | 12.3x | 6.9x | 3.6x | 19.7x | 15.7x | 0.659 | 0.069 | **0.026** | 25.3x | 2.7x | 7.2x | 3.8x | 20.5x | 16.1x |
| | 0.1% | **0.026** | 0.356 | 0.032 | 0.8x | 11.1x | 5.8x | 3.1x | 16.7x | 13.1x | 0.586 | 0.263 | **0.029** | 20.2x | 9.1x | 6.4x | 3.4x | 18.4x | 14.4x |
| | 0.5% | 0.17 | 0.338 | **0.049** | 3.5x | 6.9x | 3.8x | 2x | 10.9x | 8.5x | 1.015 | 0.372 | **0.054** | 18.8x | 6.9x | 3.4x | 1.8x | 9.9x | 7.7x |
| | 1% | 0.354 | 0.366 | **0.055** | 6.4x | 6.7x | 3.4x | 1.8x | 9.7x | 7.6x | 0.917 | 0.417 | **0.049** | 18.7x | 8.5x | 3.8x | 2x | 10.9x | 8.5x |
| WineQuality | 1 | **0.014** | 0.239 | 0.017 | 0.8x | 14.1x | 12.4x | 5.3x | 16.6x | 21.5x | 0.574 | 0.022 | **0.016** | 35.9x | 1.4x | 13.1x | 5.6x | 53.6x | 22.9x |
| | 0.1% | 0.057 | 0.262 | **0.027** | 2.1x | 9.7x | 7.8x | 3.3x | 31.8x | 13.6x | 0.329 | 0.196 | **0.024** | 13.7x | 8.2x | 8.8x | 3.8x | 35.8x | 15.3x |
| | 0.5% | 0.296 | 0.282 | **0.041** | 7.2x | 6.9x | 5.1x | 2.2x | 20.9x | 8.9x | 2.173 | 0.298 | **0.037** | 58.7x | 8.1x | 5.7x | 2.4x | 23.2x | 9.9x |
| | 1% | 0.608 | 0.276 | **0.051** | 11.9x | 5.4x | 4.1x | 1.8x | 16.8x | 7.2x | 2.711 | 0.333 | **0.051** | 53.2x | 6.5x | 4.1x | 1.8x | 16.8x | 7.2x |

**Training Time.** Table 2 shows the total training time of our framework and baselines. Our online learning framework is substantially faster than OnlineGB, DeltaBoost, and XGBoost, and slightly slower than iGBDT. While slower on smaller datasets compared to LightGBM, it outperforms on larger datasets like SUSY and HIGGS, with training times similar to MU-inGBDT. Overall, our framework offers significantly faster training than existing incr./decr. methods and is comparable to popular GBDT libraries.

Table 2: Comparison of total training time (in seconds) and memory usage (total allocated, MB).

| | Method | Adult | CreditInfo | SUSY | HIGGS | Optdigits | Pendigits | Letter | Covtype | Abalone | WineQuality |
|---|---|---|---|---|---|---|---|---|---|---|---|
| Training Time (Seconds) | iGBDT | 1.875 | 1.787 | 63.125 | 180.459 | 0.263 | 0.345 | 0.26 | 9.158 | 3.434 | 1.047 |
| | OnlineGB | 6,736.18 | 330,746.80 | OOM | OOM | 130.7 | 87.361 | 771.99 | 19,938.80 | 39.874 | 62.034 |
| | DeltaBoost | 78.213 | 154.52 | 4,281.59 | OOM | 9.517 | 18.457 | 21.532 | 582.36 | 3.104 | 4.89 |
| | MU in GBDT | 1.285 | 1.648 | 58.551 | 175.95 | 0.261 | 0.35 | 0.289 | 6.454 | 1.431 | 1.034 |
| | XGBoost | 9.467 | 13.314 | 1,634.82 | 2,230.03 | 0.752 | 0.574 | 1.171 | 63.917 | 0.186 | 0.21 |
| | LightGBM | 0.516 | 1.836 | 97.622 | 211 | 0.106 | 0.131 | 0.203 | 4.581 | 0.098 | 0.09 |
| | CatBoost | 1.532 | 3.447 | 108.95 | 303.56 | 0.177 | 0.183 | 0.232 | 6.14 | 0.533 | 0.858 |
| | ThunderGBM (GPU) | 0.564 | 0.583 | 5.993 | 13.708 | 0.296 | 0.387 | 0.366 | 1.474 | 0.418 | 0.366 |
| | Ours | 2.673 | 1.818 | 64.935 | 177.1 | 0.276 | 0.368 | 0.352 | 9.336 | 0.582 | 0.427 |
| Memory Usage (MB) | iGBDT | 1,153.13 | 2,192.13 | 31,320.40 | 31,724.40 | 2,161.20 | 3,917.61 | 3,370.38 | 18,381.10 | 1,767.23 | 1,281.08 |
| | OnlineGB | 35,804.10 | 58,119.61 | OOM | OOM | 7,493.97 | 6,488.75 | 13,067.75 | 19,699.62 | 582.97 | 345.83 |
| | DeltaBoost | 43,286.70 | 285.608 | 409,850.90 | OOM | 2,336.79 | 1,173.59 | 3,741.46 | 210,409 | 786.53 | 549.64 |
| | MU in GBDT | 570.78 | 1,095.70 | 16,576.50 | 34,380.90 | 1,080.49 | 1,959.02 | 1,805.22 | 9,637.65 | 1,711.02 | 1,194.82 |
| | XGBoost | 179.13 | 140.88 | 2,093.95 | 7,467.32 | 131.11 | 120.93 | 121.59 | 770.3 | 204.74 | 200.91 |
| | LightGBM | 150.45 | 149.19 | 1,688.57 | 4,109.54 | 121.08 | 135.45 | 161.97 | 542.47 | 215.15 | 214.95 |
| | CatBoost | 83.02 | 129.09 | 1,503.93 | 3,090.55 | 29.41 | 36.64 | 99.79 | 595.27 | 40.97 | 27.91 |
| | ThunderGBM (GPU) | 673.45 | 418.97 | 3,725.82 | 5,855.04 | 353.95 | 378.11 | 360.56 | 931.89 | 367.67 | 348.83 |
| | Ours | 577.18 | 1,096.71 | 16,576.40 | 24,333.30 | 1,081.15 | 1,959.49 | 1,805.76 | 9,665.21 | 762.78 | 531.88 |

**Memory Overhead.** Memory usage is crucial for practical applications. Most incremental and decremental learning methods store auxiliary information or learned knowledge during training, potentially occupying significant memory. As shown in Table 2, our framework's memory usage is significantly lower than OnlineGB, iGBDT, and DeltaBoost. Moreover, OnlineGB and DeltaBoost encountered OOM in the experiments.

## 4.2 ONLINE LEARNING TIME

Retraining from scratch can be time-consuming, but in some cases, the cost of online learning outweighs the benefits compared to retraining from scratch, making online learning unnecessary or unjustified. Hence, evaluating the cost of online learning is crucial for practical applications. Table 3 shows the total online learning time (seconds) and speedup v.s. baselines, comparing OnlineGB and iGBDT for incremental learning, and DeltaBoost and MUinGBDT for decremental learning.

In incremental learning, compared to OnlineGB and iGBDT, which also support incremental learning, adding a single data instance can be up to 254.4x and 17x faster, respectively. Furthermore, compared to retraining from scratch on XGBoost, LightGBM, CatBoost, and ThunderGBM (GPU), it can achieve speedups of up to 974.3x, 58.2x, 64.9x, and 27.6x, respectively. In decremental learning, when deleting a data instance, our method offers a speedup of 1,619.9x and 6.4x over DeltaBoost and MUinGBDT, respectively, and is 1,254.7x, 74.9x, 90.2x, and 29.6x faster than XGBoost, LightGBM, CatBoost, and ThunderGBM (GPU), respectively.

Our method is substantially faster than other methods both in incremental and decremental learning, especially on large datasets. For example, in HIGGS dataset, the largest dataset in our experiments, on removing (adding) 1% data, we are 3.1x faster than MUinGBDT (2.6x faster than iGBDT), while OnlineGB and DeltaBoost encounter out of memory (OOM).

Interestingly, we observed that when $|D'|$ is small, decremental learning is faster than incremental learning. However, as $|D'|$ increases, incremental learning becomes faster than decremental learning. For decremental learning, the data to be removed has already been learned, and their derivatives have been stored from training. However, the deleted data often exists discretely in memory. On the other

Table 4: The test error after training, adding, and deleting.

| Task | | Method | Adult | CreditInfo | SUSY | HIGGS | Optdigits | Pendigits | Letter | Covtype | Abalone ($\times10^{-2}$) | WineQuality ($\times10^{-3}$) |
|---|---|---|---|---|---|---|---|---|---|---|---|---|
| Training | | iGBDT | 0.1276 | 0.0629 | 0.1987 | 0.2742 | 0.0290 | 0.0295 | 0.0418 | 0.1702 | 5.7721 | 1.2085 |
| | | DeltaBoost | 0.1814 | 0.0642 | 0.2122 | OOM | 0.0652 | 0.0417 | 0.0968 | 0.2764 | 7.5905 | 1.3134 |
| | | MU in GBDT | 0.1276 | 0.0629 | 0.1987 | 0.2742 | 0.0307 | 0.0294 | 0.0418 | 0.1702 | 5.7721 | 1.2085 |
| | | XGBoost | 0.1270 | 0.0630 | 0.1977 | 0.2761 | 0.0418 | 0.0397 | 0.0524 | 0.1896 | 6.1472 | 1.1674 |
| | | LightGBM | 0.1277 | 0.0635 | 0.1984 | 0.2725 | 0.0355 | 0.0355 | 0.0374 | 0.1688 | 5.8392 | 1.1993 |
| | | CatBoost | 0.2928 | 0.1772 | 0.4324 | 0.5384 | 0.0618 | 0.0440 | 0.0655 | 0.1572 | 5.7265 | 1.2457 |
| | | ThunderGMB (GPU) | 0.2405 | 0.0659 | 0.4576 | 0.4698 | 0.2739 | 0.1155 | 0.1170 | 0.6298 | 8.4272 | 1.6953 |
| | | Ours | 0.1276 | 0.0629 | 0.1987 | 0.2742 | 0.0307 | 0.0294 | 0.0418 | 0.1702 | 5.7721 | 1.2085 |
| Incre. Learning | Add 1 | iGBDT | 0.1279 | 0.0633 | **0.1987** | 0.2769 | 0.0301 | **0.0286** | 0.0418 | 0.1696 | 5.8801 | **1.1953** |
| | | Ours | **0.1275** | **0.0630** | 0.1988 | **0.2742** | **0.0295** | 0.0297 | **0.0404** | **0.1685** | **5.811** | 1.2079 |
| | Add 0.1% | iGBDT | 0.1267 | 0.0630 | 0.1995 | **0.2742** | 0.0323 | 0.0363 | 0.0446 | 0.1777 | 6.2531 | 1.2680 |
| | | Ours | 0.1269 | **0.0626** | 0.1989 | 0.2747 | **0.0295** | 0.0297 | 0.0406 | 0.1686 | 5.900 | 1.2040 |
| | Add 0.5% | iGBDT | **0.1287** | 0.0636 | 0.2012 | 0.2795 | 0.0390 | 0.0440 | 0.0572 | 0.1788 | 7.6510 | 1.2907 |
| | | Ours | 0.1294 | **0.0632** | 0.1988 | 0.2734 | **0.0290** | 0.0295 | 0.0394 | 0.1681 | 5.7701 | 1.2198 |
| | Add 1% | iGBDT | 0.1291 | **0.0630** | 0.2014 | 0.2780 | 0.0529 | 0.0603 | 0.0875 | 0.1868 | 8.5324 | 1.4462 |
| | | Ours | **0.1267** | 0.0632 | **0.1990** | 0.2740 | **0.0262** | 0.0283 | 0.0440 | 0.1683 | **5.8378** | **1.2209** |
| Decre. Learning | Del 1 | DeltaBoost | 0.1818 | 0.0642 | 0.2122 | OOM | 0.0640 | 0.0424 | 0.0974 | 0.2764 | 7.4359 | 1.3084 |
| | | MU in GBDT | **0.1280** | 0.0629 | **0.1987** | **0.2742** | **0.0306** | **0.0295** | **0.0408** | **0.1702** | 5.8025 | **1.2095** |
| | | Ours | 0.1276 | **0.0628** | **0.1987** | **0.2742** | **0.0306** | **0.0295** | 0.0416 | **0.1702** | 5.8723 | 1.2143 |
| | Del 0.1% | DeltaBoost | 0.1823 | 0.066 | 0.2122 | OOM | 0.0629 | 0.0412 | 0.0956 | 0.2764 | 7.3402 | 1.3159 |
| | | MU in GBDT | 0.1285 | **0.0634** | 0.1988 | **0.2742** | 0.0301 | 0.0295 | 0.0444 | 0.1734 | 5.9727 | 1.2202 |
| | | Ours | **0.1284** | 0.0633 | 0.1988 | 0.2747 | **0.0295** | **0.0283** | **0.0432** | 0.1712 | **5.8744** | **1.2109** |
| | Del 0.5% | DeltaBoost | 0.1829 | 0.0642 | 0.2122 | OOM | 0.0663 | 0.0423 | 0.0960 | 0.2762 | 7.2955 | 1.3022 |
| | | MU in GBDT | 0.1309 | 0.0640 | 0.1988 | 0.2751 | 0.0306 | **0.0283** | 0.0442 | 0.1727 | 6.3142 | 1.2398 |
| | | Ours | **0.1295** | **0.0634** | 0.1988 | 0.2746 | **0.0301** | 0.0303 | **0.0432** | **0.1675** | **5.7733** | **1.2052** |
| | Del 1% | DeltaBoost | 0.1812 | 0.0642 | 0.2123 | OOM | 0.0624 | 0.0435 | 0.0958 | 0.2764 | 7.3100 | 1.3163 |
| | | MU in GBDT | 0.1311 | 0.0639 | 0.1988 | **0.2745** | 0.0334 | 0.0312 | 0.0460 | 0.1766 | 6.3558 | 1.2925 |
| | | Ours | **0.1295** | **0.0632** | **0.1987** | 0.2747 | **0.0273** | **0.0303** | **0.0424** | **0.1695** | **5.7620** | **1.2111** |

hand, for incremental learning, the data to be added are unseen, and derivatives need to be computed during the incremental learning process. Nevertheless, we append the added data at the end, ensuring that the added data are stored contiguously in memory. With a small $|D'|$, derivatives can be reused in decremental learning, whereas derivatives need to be computed in incremental learning. Therefore, decremental learning is less time-consuming. However, as $|D'|$ grows, continuous memory access in incremental learning is faster than decremental learning, making incremental learning faster.

## 4.3 TEST ERROR RATE

Table 4 presents the test error for different methods, defined as (1 - accuracy) for classification tasks and Mean Squared Error (MSE) for regression tasks. Due to page limitations, we have omitted the results for OnlineGB, as its excessively long learning time makes it relatively insignificant compared to the other methods. Three scenarios are considered: (1) Training, reporting the test error for models trained on the full dataset $D$; (2) Incremental Learning, performing incremental learning to add a randomly selected portion $D'$ into a model pre-trained on $D - D'$; and (3) Decremental Learning, conducting decremental learning to remove $D'$ from a model trained on the full dataset $D$. Our method achieved the best error rates in most cases.

## 4.4 BATCH ADDITION & REMOVAL

In the traditional setting, GBDT models must be trained in one step with access to all training data, and they cannot be modified after training – data cannot be added or removed. In our proposed online learning framework, GBDT models support both incremental and decremental learning, allowing continual batch learning (data addition) and batch removal, similar to mini-batch learning in DNNs.

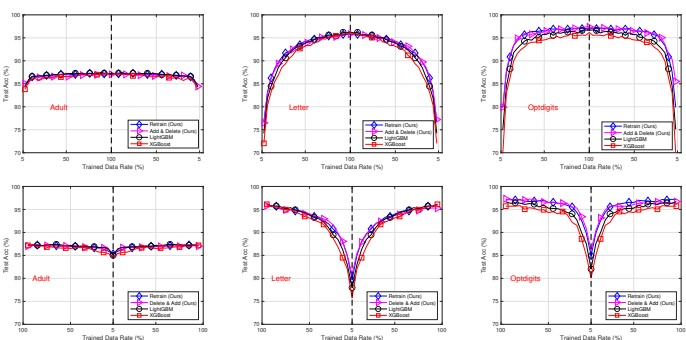

Figure 3: The impact of tuning data size on the number of retrained nodes for each iteration in incremental learning.

We conducted experiments on continual batch addition and removal by dividing the data into 20 equal parts, each with $5\%|D_{\mathrm{tr}}|$. Figure 3 (top) shows a GBDT model incrementally trained from $5\%$ to $100\%$ of the data, then decrementally reduced back to $5\%$. We retrained models for comparison. Figure 3 (bottom) depicts a model decrementally reduced from $100\%$ to $5\%$, then incrementally trained back to $100\%$. We also report the accuracy of XGBoost and LightGBM. The overlapping curves demonstrate the effectiveness of our online learning framework. Due to space limitations, results are shown for only three datasets.

## 4.5 DATA ADDITION WITH MORE CLASSES

Our framework can update data with unseen classes. We divide the dataset into sub-datasets based on labels (e.g., Optdigits has 10 labels, so we divide it into 10 sub-datasets). We train a model on the first sub-dataset and test it on two test datasets: 1) the original full test dataset with all labels, and 2)

the partial test dataset with only the learned labels. We fine-tune the model with a new sub-dataset through incremental learning until learning the full dataset, testing the model on both test datasets after each training. Figure 4 shows that the accuracy of incremental learning and retraining is nearly identical on both the full and partial datasets. Note that the decrease in accuracy on the partial dataset is likely due to the increasing complexity of the learned data, which leads to a decrease in accuracy.

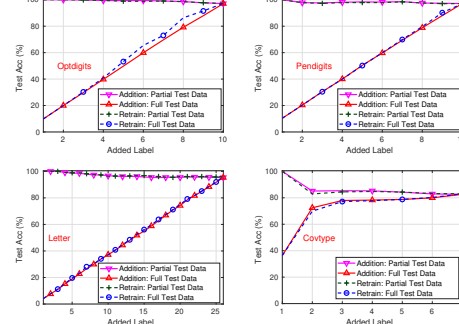

Figure 4: The impact of tuning data size on the number of retrained nodes for each iteration in incremental learning.

### 4.6 Verifying by Backdoor Attacking

Backdoor attacks in machine learning refers to a type of malicious manipulation of a trained model, which is designed to modify the model's behavior or output when it encounters a specific, predefined trigger input pattern Salem et al. (2022); Saha et al. (2020). In this evaluation, we demonstrate that our framework can successfully inject and remove backdoor in a well-trained, clean GBDT model using incremental learning and decremental learning. The details of backdoor attack experiments are provided in Appendix G.

In this evaluation, we randomly selected a subset of the training dataset and injected triggers into it to create a backdoor training dataset, leaving the rest as the clean training dataset. The test dataset was similarly divided into backdoor and clean subsets. We report the accuracy for clean

Table 5: Accuracy for clean test dataset and attack successful rate for backdoor test dataset.

| Dataset | Train Clean | | Train Backdoor | | Add Backdoor | | Remove Backdoor | |
|---|---|---|---|---|---|---|---|---|
| | Clean | Backdoor | Clean | Backdoor | Clean | Backdoor | Clean | Backdoor |
| Optdigits | 96.21% | 8.91% | 96.27% | 100% | 95.94% | 100% | 95.82% | 9.69% |
| Pendigits | 96.11% | 3.97% | 96.43% | 100% | 96.48% | 100% | 96.51% | 5.55% |
| Letter | 93.9% | 1.38% | 94.08% | 100% | 93.62% | 100% | 93.78% | 3.48% |
| Covtype | 78.4% | 47.83% | 78.32% | 100% | 78.38% | 100% | 78.38% | 51.71% |

test dataset and attack successful rate (ASR) for backdoor test dataset in Table 5. Initially, we trained a model on the clean training data ("Train Clean"), which achieved high accuracy on the clean test dataset but low ASR on the backdoor test dataset. We then incrementally add the backdoor training data with triggers in to the model ("Add Backdoor"). After incremental learning, the model attained 100% ASR on the backdoor test dataset, demonstrating effective learning of the backdoor data. For comparison, training a model on the combined clean and backdoor training datasets ("Train Backdoor") yielded similar results to "Add Backdoor". Finally, we removed the backdoor data using decremental learning ("Remove Backdoor"), reducing the ASR to the level of the clean model and confirming the successful removal of backdoor data. This evaluation shows the effectiveness of our online learning framework in handling backdoor attacks.

### 4.7 Verifying by Membership Inference Attack

The membership inference attack (MIA) aims to predict whether a data sample is part of the training dataset Shokri et al. (2017); Hu et al. (2022); Choquette-Choo et al. (2021). Therefore, the goal of this experiments is to determine if "deleted" data can still be identified as training data after decremental learning. However, in our experiment with default hyper-parameter setting, the predictions made by MIA are nearly random guesses.

**Experimental Setup.** Previous studies demonstrate that overfitting can make machine learning models more vulnerable to MIA Yeom et al. (2018); van Breugel et al. (2023); Hu et al. (2022). To further validate our approach, we apply a smaller model with number of iteration $M = 5$, which can be easily overfitted. Conducting MIA on such a small model can further confirm our approach indeed deletes the data from the model. Therefore, we conduct the this experiments on multi-class datasets: Optdigits, Pendigits, Letter and Covtype. For each dataset, we split it into three subsets: base dataset $D_{\text{base}}$, online dataset $D'$, and test dataset $D_{\text{test}}$. We first train a base model on $D_{\text{base}} + D'$. For this base model, the MIA should identify the data in $D'$ as part of the training dataset. Next, we perform decremental learning to delete $D'$ from the base model. After this process, the MIA should no longer identify the data in $D'$ as part of the training dataset. Finally, we add $D'$ back to the model using incremental learning. Following this, the MIA should once again identify the data in $D'$ as part of the training dataset. Further details about the MIA experiment are provided in Appendix H.

**Results.** Table 6 presents the average probability of data samples being identified as part of the training dataset at different stages. For the base model, MIA identifies 100% of the data in $D_{\text{base}}$ and $D'$ as part of the training dataset, while the data in $D_{\text{test}}$ has a low probability of being identified as

part of the training dataset. After decremental learning, the probability for $D_{base}$ remains unchanged, while the probability for $D'$ drops to a level almost identical to $D_{test}$. This confirms that $D'$ has been effectively deleted from the

Table 6: Membership Inference Attack.

| Dataset | Base Model | | | After decremetal learning | | | After incremetal learning | | |
|---|---|---|---|---|---|---|---|---|---|
| | $D_{base}$ | $D'$ | $D_{test}$ | $D_{base}$ | $D'$ | $D_{test}$ | $D_{base}$ | $D'$ | $D_{test}$ |
| Optdigits | 100% | 100% | 43.59% | 100% | 33.93% | 42.19% | 100% | 100% | 43.82% |
| Pendigits | 100% | 100% | 56.09% | 100% | 55.04% | 46.15% | 100% | 100% | 56.63% |
| Letter | 100% | 100% | 26.31% | 100% | 13.33% | 47.37% | 100% | 100% | 36.84% |
| Covtype | 100% | 100% | 38.89% | 100% | 15.2% | 38.89% | 100% | 100% | 44.31% |

base model. After incremental learning, the probability for $D'$ increases to 100% again, indicating that the model has relearned $D'$. The probability for $D_{test}$ in the incremental model remains almost the same as in the base model. This result confirms that our decremental/incremental learning approach can indeed delete/add data from/to the model.

### 4.8 ADDITIONAL EVALUATIONS

To further verify our method's effectiveness and efficiency, we provide additional evaluations:

• **Extremely High-dimensional Datasets:** To confirm the scalability of our framework, we report the experiments for two extremely high-dimensional datasets, RCV1 and News20, in Appendix I.

• **Model Functional Similarity:** We report this metric in Appendix F to evaluate the similarity between the model learned by online learning and the one retrained from scratch.

• **Approximation Error of Leaf Scores:** Since the outdated derivatives are used in gain computation, to evaluate the effect of these outdated derivatives, we report the approximation error of leaf scores between the model after addition/deletion and the one retrained from scratch in Appendix J.

• **Ablation Study:** We report the ablation study for different hyper-parameter settings in Appendix K.

## 5 RELATED WORK

**Incremental Learning** is a technique in machine learning that involves the gradual integration of new data into an existing model, continuously learning from the latest data to ensure performance on new data van de Ven et al. (2022). It has been a open problem in machine learning, and has been studied in convolutional neural network (CNN) Polikar et al. (2001); Kuzborskij et al. (2013); Zhou et al. (2022), DNN Hussain et al. (2023); Dekhovich et al. (2023), SVM Chen et al. (2019); Cauwenberghs & Poggio (2000) , random forest (RF) Wang et al. (2009); Brophy & Lowd (2020). In gradient boosting, iGBDT Zhang et al. (2019) offers incremental updates, while other methods Beygelzimer et al. (2015a); Babenko et al. (2009) extend gradient boosting to online learning. However, these methods do not support removing data.

**Decremental Learning** allows for the removal of trained data and eliminates their influence on the model, which can be used to delete outdated or privacy-sensitive data Bourtoule et al. (2021); Nguyen et al. (2022); Sekhari et al. (2021); Xu et al. (2024). It has been researched in various models, including CNN Poppi et al. (2023); Tarun et al. (2021), DNN Chen et al. (2023); Thudi et al. (2022), SVM Karasuyama & Takeuchi (2009); Cauwenberghs & Poggio (2000), Naive Bayes Cao & Yang (2015), K-means Ginart et al. (2019), RF Schelter et al. (2021); Brophy & Lowd (2021), and gradient boosting Wu et al. (2023); Zhang et al. (2023). In random forests, DaRE Brophy & Lowd (2021) and a decremental learning algorithm Schelter et al. (2021) were proposed for data removal with minimal retraining and latency.

However, in GBDT, trees in subsequent iterations rely on residuals from previous iterations, making decremental learning more complicated. DeltaBoost Wu et al. (2023) simplified the dependency for data deletion by dividing the dataset into disjoint sub-datasets, while a recent study Lin et al. (2023) proposed an efficient unlearning framework without simplification, utilizing auxiliary information to reduce unlearning time. Although effective, its performance on large datasets remains unsatisfactory.

## 6 CONCLUSION

In this paper, we propose an novel in-place online learning framework for GBDT that support incremental and decremental learning: it enables us to dynamically add a new dataset to the model and delete a learned dataset from the model. It support continual batch addition/removal, and data additional with unseen classes. We present a collection of optimizations on our framework to reduce the cost of online learning. Adding or deleting a small fraction of data is substantially faster than retraining from scratch. Our experimental results, including backdoor attack, membership inference attack, and other empirical evaluations confirm the effectiveness and efficiency of our framework and optimizations – successfully adding or deleting data while maintaining accuracy.

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

## A  FEATURE DISCRETIZATION.

The preprocessing step of feature discretization plays a crucial role in simplifying the implementation of Eq. equation 5 and reducing the number of splits that need to be evaluated. This process involves sorting the data points based on their feature values and assigning them to bins, taking into account the distribution of the data, as shown in Figure 5 and Algorithm 4. By starting with a small bin-width (e.g., $10^{-8}$) and a predetermined maximum number of bins $B$ (e.g., 1024). It assigns bin numbers to the data points from the smallest to the largest, carefully considering

the presence of data points in each bin. This iterative process continues until the number of bins exceeds the specified maximum.

In cases where the number of required bins surpasses the maximum limit, the bin-width is doubled, and the entire process is repeated. This adaptive discretization approach proves particularly effective for boosted tree methods, ensuring that feature values are mapped to integers within a specific range. Consequently, after the discretization mapping is established, each feature value is assigned to the nearest bin. After this discretization preprocessing, all feature values are integers within $\{0, 1, 2, \cdots, B-1\}$.

The advantage of this discretization technique becomes evident during the gain searching step. Instead of iterating over all $N$ feature values, the algorithm only needs to consider a maximum of $B$ splits for each feature. This substantial reduction in the number of splits to evaluate leads to a significant decrease in the computational cost, transforming it from being dependent on the dataset size $N$ to a manageable constant $B$.

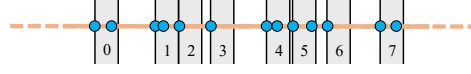

Figure 5: Feature discretization example. For a feature, all its values are grouped into 8 bins, i.e., the original feature values become integers between 0 to 7 assigned to the nearest bin.

---

**Algorithm 4** Discretize Feature

1: $v_{\{1..N\}}$ = sorted feature values, $bin\_width = 10^{-10}$
2: **while true do**
3:    $cnt = 0$, $curr\_idx = 0$
4:    **for** $i = 1$ **to** $N$ **do**
5:       **if** $v_i - v_{curr\_idx} > bin\_width$ **then**
6:          $cnt = cnt + 1$, $cur\_idx = i$
7:          **if** $cnt > B$ **then**
8:             $bin\_width = bin\_width * 2$
9:             **break**
10:          **end if**
11:       **end if**
12:       $v_i' = cnt$
13:    **end for**
14:    **if** $cnt <= B$ **then break**
15: **end while**
16: **return** $v'$ as discretized feature values

---

## B  Experiment Setting

The experiments are performed on a Linux computing node running Red Hat Enterprise Linux 7, utilizing kernel version 5.10.155-1.el7.x86_64. The CPU employed was an Intel(R) Xeon(R) Gold 6150 CPU operating at a clock speed of 2.70GHz, featuring 18 cores and 36 threads. The system was equipped with a total memory capacity of 376 GB. We have built a prototype of our online learning framework using C++11. The code is compiled with g++-11.2.0, utilizing the "O3" optimization. Unless explicitly stated otherwise, our default parameter settings are as follows: $J = 20$, $B = 1024$, $|D'| = 0.1\% \times |D_{tr}|$, $\alpha = 0.1$, and $\sigma = 0.1$. We report the ablation study for different settings in Appendix K.

## C  Framework Overview

Figure 1 is a visual example of incremental and decremental learning of our proposed framework. Figure 1(b) is one tree of the GBDT model and has been well-trained on dataset $D_{tr} = \{0, 1, 2, 3..., 19\}$. Every rectangle in the tree represents a node, and the labels inside indicate the splitting criteria. For instance, if the condition `Age < 42` is met, the left-child node is followed; otherwise, the right-child node is chosen. The numbers within the rectangles represent the prediction value of the terminal nodes. Please note that here the feature `42` is a discretized value, instead of the raw feature. Our online learning framework has the capability to not only incrementally learn a new dataset $D_{in}$, but also decrementally delete a learned dataset $D_{de} \subseteq D_{tr}$.

**Example for Incremental Learning.** Here, we would like to add a new dataset $D' = D_{in} = \{20, 21, 22, 23\}$ to the original model, so we will call the function of incremental learning. $|d|$ denotes how many data of the $D'$ reach this node. As shown in Algorithm 3, we traverse all non-terminal nodes (non-leaf nodes) in the tree at first. For example, we are going to test the node of `Loan < 31`. Its current best split is `Loan < 31`. One of the new data instances $\{22\}$ reaches this node. After adding this data and recomputing the gain value, `Loan < 31` is still best split with the greatest gain value of 26.937, and meets $s = s'$, as shown in Figure 1(a). Thus, we can keep this split and do not need to do any changes for this node. Then we are going to test the node of `Auto < 57` and the remaining three new data instances $\{20, 21, 23\}$ reach this node. As shown in the left side of Figure 1(c), we recompute the gain value for this node, but the best split changes to `Income < 5`. Therefore, we retrain the pending sub-tree rooted on `Auto < 57` after adding new data instances to

obtain a new sub-tree rooted on `Income < 5`. Then we replace the pending sub-tree with the new one. Finally, we update the prediction value on terminal nodes (leaf nodes). For example, $0.4322$ is updated to $0.2735$ because of adding data $\{22\}$; $-0.1252$ has no change because the data of this node are still the same.

**Example for Decremental Learning.** Similar to incremental learning, we would like to delete a learned dataset $D_{de} = \{2, 7, 11, 13\}$ and its effect on the model. The best split of node `Loan < 31` does not change, so we keep the split. For `Auto < 57`, as shown in the right side of Figure 1(c), after removing data instances $\{2, 11, 13\}$, the best split changes from `Auto < 57` to `Credit < 24`, so we retrain the pending sub-tree rooted on `Loan < 31` and then replace it with the new sub-tree. For terminal nodes (leaf nodes), the prediction value changes if any data reaching this node is removed.

# D  SPLIT CANDIDATES SAMPLING

**Definition 1** (Distance Robust) *Let $s$ be the best split, and $\frac{|D'|}{|D_{tr}|} = \lambda$. $N_\Delta$ is the distance between $s$ and its nearest split $t$, $N_\Delta = ||t - s||$. $s$ is distance robust if*

$$N_\Delta > \frac{\lambda Gain(s)}{\frac{1}{N_{ls}} \frac{\left(\sum_{\mathbf{x}_i \in l_s} g_{i,k}\right)^2}{\sum_{\mathbf{x}_i \in l_s} h_{i,k}} + \frac{1}{N_{rs}} \frac{\left(\sum_{\mathbf{x}_i \in r_s} g_{i,k}\right)^2}{\sum_{\mathbf{x}_i \in r_s} h_{i,k}}} \tag{8}$$

*Proof.* In decremental learning, for a fixed $\lambda$, we have

$$(1 - \lambda)\, Gain(s) - Gain(s + N_\Delta) \tag{9}$$

$$\approx (1 - \lambda)\left(\frac{\left(\sum_{\mathbf{x}_i \in l_s} g_{i,k}\right)^2}{\sum_{\mathbf{x}_i \in l_s} h_{i,k}} + \frac{\left(\sum_{\mathbf{x}_i \in r_s} g_{i,k}\right)^2}{\sum_{\mathbf{x}_i \in r_s} h_{i,k}} - \frac{\left(\sum_{\mathbf{x}_i \in l_s \cup r_s} g_{i,k}\right)^2}{\sum_{\mathbf{x}_i \in l_s \cup r_s} h_{i,k}}\right)$$

$$- \left(\left(1 - \frac{N_\Delta}{N_{ls}}\right) \frac{\left(\sum_{\mathbf{x}_i \in l_s} g_{i,k}\right)^2}{\sum_{\mathbf{x}_i \in l_s} h_{i,k}} + \left(1 - \frac{N_\Delta}{N_{rs}}\right) \frac{\left(\sum_{\mathbf{x}_i \in r_s} g_{i,k}\right)^2}{\sum_{\mathbf{x}_i \in r_s} h_{i,k}}\right.$$

$$\left. - \frac{\left(\sum_{\mathbf{x}_i \in l_s \cup r_s} g_{i,k}\right)^2}{\sum_{\mathbf{x}_i \in l_s \cup r_s} h_{i,k}}\right) \tag{10}$$

where $l$ represents the left child of split $s$, and it contains the samples belonging to this node, while $r$ represent the right child, $N_{ls}$ denotes $|l_s|$, and $N_{rs}$ denotes $|r_s|$.

Let $(1 - \lambda)Gain(s) - Gain(s + N_\Delta) > 0$, we have

$$\overset{approx}{\Rightarrow} (1 - \lambda)Gain(s) - \left(\left(1 + \frac{N_\Delta}{N_{ls}}\right) \frac{\left(\sum_{\mathbf{x}_i \in l_s} g_{i,k}\right)^2}{\sum_{\mathbf{x}_i \in r_s} h_{i,k}}\right.$$

$$\left. + \left(1 - \frac{N_\Delta}{N_{rs}}\right) \frac{\left(\sum_{\mathbf{x}_i \in r_s} g_{i,k}\right)^2}{\sum_{\mathbf{x}_i \in r_s} h_{i,k}} - \frac{\left(\sum_{\mathbf{x}_i \in l_s \cup r_s} g_{i,k}\right)^2}{\sum_{\mathbf{x}_i \in l_s \cup r_s} h_{i,k}}\right) \tag{11}$$

$$\Rightarrow \frac{N_\Delta}{N_{ls}} \frac{\left(\sum_{\mathbf{x}_i \in l_s} g_{i,k}\right)^2}{\sum_{\mathbf{x}_i \in l_s} h_{i,k}} + \frac{N_\Delta}{N_{rs}} \frac{\left(\sum_{\mathbf{x}_i \in r_s} g_{i,k}\right)^2}{\sum_{\mathbf{x}_i \in r_s} h_{i,k}} - \lambda Gain(s) > 0 \tag{12}$$

$$\Rightarrow N_\Delta > \frac{\lambda Gain(s)}{\frac{1}{N_{ls}} \frac{\left(\sum_{\mathbf{x}_i \in l_s} g_{i,k}\right)^2}{\sum_{\mathbf{x}_i \in l_s} h_{i,k}} + \frac{1}{N_{rs}} \frac{\left(\sum_{\mathbf{x}_i \in r_s} g_{i,k}\right)^2}{\sum_{\mathbf{x}_i \in r_s} h_{i,k}}} \tag{13}$$

$\square$

In the above definition, $\mathbb{E}(N_\Delta) = 1/\alpha$, where $\alpha$ denotes the split sampling rate, we can observe that a smaller sampling rate will result in a more robust split, so we can reduce the number of retrain operations by reducing the sampling rate. Similarly, incremental learning can get the same result.

**Definition 2** (Robustness Split) *For a best split $s$ and an split $t$ with the same feature, $t \neq s$, and online learning data rate $\frac{|D'|}{|D_{tr}|} = \lambda$, the best split $s$ is robust split if*

$$Gain(s) > \frac{1}{1 - \lambda} Gain(t) \tag{14}$$

*Proof.* Initially, we have

$$Gain(s) = \frac{\left(\sum_{\mathbf{x}_i \in l_s} g_{i,k}\right)^2}{\sum_{\mathbf{x}_i \in l_s} h_{i,k}} + \frac{\left(\sum_{\mathbf{x}_i \in r_s} g_{i,k}\right)^2}{\sum_{\mathbf{x}_i \in r_s} h_{i,k}} - \frac{\left(\sum_{\mathbf{x}_i \in l_s \cup r_s} g_{i,k}\right)^2}{\sum_{\mathbf{x}_i \in l_s \cup r_s} h_{i,k}} \tag{15}$$

After decremental learning, we get

$$Gain'(s) = \frac{\left(\sum_{\mathbf{x}_i \in l_s} g_{i,k} - \sum_{\mathbf{x}_i \in l_s \cap D'} g_{i,k}\right)^2}{\sum_{\mathbf{x}_i \in l_s} h_{i,k} - \sum_{\mathbf{x}_i \in l_s \cap D'} h_{i,k}} + \frac{\left(\sum_{\mathbf{x}_i \in r_s} g_{i,k} - \sum_{\mathbf{x}_i \in r_s \cap D'} g_{i,k}\right)^2}{\sum_{\mathbf{x}_i \in r_s} h_{i,k} \sum_{\mathbf{x}_i \in r_s \cap D'} h_{i,k}}$$

$$- \frac{\left(\sum_{\mathbf{x}_i \in l_s \cup r_s} g_{i,k} - \sum_{\mathbf{x}_i \in (l_s \cup r_s) \cap D'} g_{i,k}\right)^2}{\sum_{\mathbf{x}_i \in l_s \cup r_s} h_{i,k} - \sum_{\mathbf{x}_i \in (l_s \cup r_s) \cap D'} h_{i,k}} \tag{16}$$

For any possible split $t$ ($t \neq s$), the split $s$ is robust only and only if $Gain(s) > Gain(t)$ and $Gain'(s) > Gain'(t)$. First, let's analyze the first term of $Gain'(s)$. Suppose $\frac{|D'|}{|D_{tr}|} = \lambda$, and $D'$ is randomly selected from $D$. Here we consider the leaf child $l_s$ of split $s$, and let the $|l_s \cap D'|$ to be $n_{ls}$, $|l_s|$ to be $N_{ls}$. Then we have

$$\frac{\left(\sum_{\mathbf{x}_i \in l_s} g_{i,k} - \sum_{\mathbf{x}_i \in l_s \cap D'} g_{i,k}\right)^2}{\sum_{\mathbf{x}_i \in l_s} h_{i,k} - \sum_{\mathbf{x}_i \in l_s \cap D'} h_{i,k}} \overset{approx}{\Longrightarrow} \frac{\left(\sum_{\mathbf{x}_i \in l_s} g_{i,k} - n_{ls} \overline{g}_{ls}\right)^2}{\sum_{\mathbf{x}_i \in l_s} h_{i,k} - n_{ls} \overline{h}_{ls}} \tag{17}$$

$$\Rightarrow \left(1 - \frac{n_{ls}}{N_{ls}}\right) \frac{\left(\sum_{\mathbf{x}_i \in l_s} g_{i,k}\right)^2}{\sum_{\mathbf{x}_i \in l_s} h_{i,k}} \tag{18}$$

where $\overline{g}$ and $\overline{h}$ denote the average of the $g_{i,k}$ and $h_{i,k}$ respectively.

Similarly, we can get all three terms for $Gain(s)$, $Gain'(s)$, $Gain(t)$, and $Gain'(t)$ in a similar form. For $Gain'(s) > Gain'(t)$, finally, we have $Gain(s) > Gain(t) + C$, where

$$C = \left( \frac{n_{ls}}{N_{ls}} \frac{\left(\sum_{\mathbf{x}_i \in l_s} g_{i,k}\right)^2}{\sum_{\mathbf{x}_i \in r_s} h_{i,k}} + \frac{n_{rs}}{N_{rs}} \frac{\left(\sum_{\mathbf{x}_i \in r_s} g_{i,k}\right)^2}{\sum_{\mathbf{x}_i \in r_s} h_{i,k}} \right.$$

$$\left. - \frac{n_{ls} + n_{rs}}{N_{ls} + N_{rs}} \frac{\left(\sum_{\mathbf{x}_i \in l_s \cup r_s} g_{i,k}\right)^2}{\sum_{\mathbf{x}_i \in l_s \cup r_s} h_{i,k}} \right) - \left( \frac{n_{lt}}{N_{lt}} \frac{\left(\sum_{\mathbf{x}_i \in l_t} g_{i,k}\right)^2}{\sum_{\mathbf{x}_i \in r_t} h_{i,k}} \right.$$

$$\left. + \frac{n_{rt}}{N_{rt}} \frac{\left(\sum_{\mathbf{x}_i \in r_t} g_{i,k}\right)^2}{\sum_{\mathbf{x}_i \in r_t} h_{i,k}} - \frac{n_{lt} + n_{rt}}{N_{lt} + N_{rt}} \frac{\left(\sum_{\mathbf{x}_i \in l_t \cup r_t} g_{i,k}\right)^2}{\sum_{\mathbf{x}_i \in l_t \cup r_t} h_{i,k}} \right) \tag{19}$$

The upper bound of $C$ is $\lambda Gain(s)$. Further, we have

$$Gain(s) > \frac{1}{1 - \lambda} Gain(t) \tag{20}$$

$\square$

The above definition shows that, as $\lambda = \frac{|D'|}{|D_{tr}|}$ decreases, the splits are more robust, leading to a reduction in the frequency of retraining. In conclusion, decreasing either $\alpha$ or $\lambda$ makes the split more robust, reducing the change occurrence in the best split, and it can significantly reduce the online learning time.

## E  Update w/o Touching Training Data

**Maintain Best Split.** The split gain is calculated by Eq. equation 5. There are three terms: the gain for the left-child, the gain for the right-child, and subtracting the gain before the split. Each gain is computed as the sum of the squared first derivatives $\left(\left[\sum_{i=1}^{N}\left(r_{i,k}-p_{i,k}\right)\right]^2\right)$ divided by the sum of the second derivatives $\left(\sum_{i=1}^{N}p_{i,k}(1-p_{i,k})\right)$ for all the data in the node. To compute these terms, it is necessary to iterate over all the data that reaches the current node. The most straightforward way for online learning to obtain the split gain is to directly compute these three terms for dataset $D_{tr} \pm D'$. In the worst case, which is the root node, the computation cost for gain computing is $|D_{tr}| + |D_{in}|$ or $|D_{tr}| - |D_{de}|$ because the root node contains all the training data.

We calculate the split gain for $D_{tr} \pm D'$ without touching the $D_{tr}$. In this optimization, during the training process, we store the $S_{rp} = \sum_{i=1}^{N}\left(r_{i,k}-p_{i,k}\right)$ and $S_{pp} = \sum_{i=1}^{N}p_{i,k}(1-p_{i,k})$ for the training dataset $D_{tr}$ for every potential split. In incremental learning process, we can only calculate the $S'_{rp}$ and $S'_{pp}$ for $D_{in}$. To obtain the new split gain based on Eq. equation 5, we add it to the stored $S_{rp}$ and $S_{pp}$. Similarly, for decremental learning, we can only calculate the $S'_{rp}$ and $S'_{pp}$ for $D_{de}$ to obtain the new split gain. In this manner, we successfully avoid the original training data for split gain computation and reduce the computation cost from $O(D_{tr} \pm D')$ to $O(D')$.

**Recomputing Prediction Value.** For the terminal node (leaf node), if there are no data of $D'$ reaching this node, we can skip this node and do not need to change the prediction value. Otherwise, we have to calculate a prediction value $f$ as shown in line 5 of the Algorithm 1. Similar to split gain computing, it is required to iterate over all the data that reaches this terminal node. Here we store $S_{rp} = \sum_{\mathbf{x}_i \in R_{j,k,m}}(r_{i,k}-p_{i,k})$ and $S_{pp} = \sum_{\mathbf{x}_i \in R_{j,k,m}}(1-p_{i,k})\,p_{i,k}$ for training dataset $D_{tr}$ in training process. Thus, in online learning process, we only need to calculate $S'_{rp}$ and $S'_{pp}$ for online learning dataset $D'$.

**Incremental Update for Derivatives.** After conducting online learning on a tree, we need to update the derivatives and residuals for learning the next tree. From the perspective of GBDT training, each tree in the ensemble is built using the residuals learned from the trees constructed in all previous iterations: Modifying one of the trees affects all the subsequent trees. A trivial method is to update the derivatives and residuals for all data instances of $D_{tr} \pm D'$ in every tree, but it is time-consuming.

When performing online learning on a tree, not all terminal nodes will be changed—some terminal nodes remain unchanged because there is no data from $D'$ that reaches these terminal nodes. Note that our goal is to find a model close to the model retraining from scratch. In the online learning scenario, all trees have already been well-trained on $D_{tr}$. Intuitively, the derivative changes for data in those unchanged terminal nodes should be minimal. Therefore, as shown in Figure 1(d), we only update the derivatives for those data reaching the changed terminal nodes. For example, the terminal node with a prediction value of $-0.1252$ does not meet any data in $D'$ in both incremental learning and decremental learning, so the prediction value of this node does not need to be changed. Therefore, we do not need to update the derivatives of the data $\{1, 6, 14, 16, 17\}$ reaching this terminal node.

## F  Model Functional Similarity

As mentioned in Section 2.2, the goal of the framework is to find a model close to the model retrained from scratch. The model functional similarity is a metric to evaluate how close the model learned by online learning and the one retrained from scratch. We show the model functional similarity for incremental learning and decremental learning in Table 7. C2W refers to the ratio of testing instances that are correctly predicted during retraining but are wrongly predicted after decremental learning. Similarly, W2C represents the testing instances that are wrongly predicted during retraining but are correctly predicted after decremental learning. The W2W column indicates the cases where the two models have different wrong predictions. For binary labels, W2W is not applicable. In the $|D'|$ column, 1 indicates that only add/remove one instance, while 0.1% corresponds to $|D'| = 0.1\% \times |D_{tr}|$. We present $\phi$ to evaluate the model functional similarity (adapted from the model functionality Adi et al. (2018)), indicating the leakage of online learning:

Table 7: Model functionality change after online learning.

| Dataset | Metric | iGBDT (Incr.) | | Ours (Incr.) | | DeltaBoost (Decr.) | | MUinGBDT (Decr.) | | Ours (Decr.) | |
|---------|--------|-------|---------|-------|---------|-------|---------|-------|---------|-------|---------|
| | | Add 1 | Add 0.1% | Add 1 | Add 0.1% | Del 1 | Del 0.1% | Del 1 | Del 0.1% | Del 1 | Del 0.1% |
| Adult | C2W ↓ | 0.40% | 0.93% | 0.17% | 0.61% | 1.17% | 1.87% | 0.63% | 0.51% | 0.55% | 0.51% |
| | W2C ↓ | 0.27% | 0.80% | 0.18% | 0.56% | 0.72% | 1.28% | 0.60% | 0.73% | 0.56% | 0.68% |
| | φ ↑ | 99.34% | 98.27% | 99.66% | 98.83% | 98.11% | 96.85% | 98.77% | 98.76% | 98.88% | 98.82% |
| CreditInfo | C2W ↓ | 0.21% | 0.40% | 0.16% | 0.30% | 0.58% | 0.92% | 0.10% | 0.21% | 0.10% | 0.18% |
| | W2C ↓ | 0.18% | 0.40% | 0.15% | 0.29% | 0.08% | 0.13% | 0.08% | 0.23% | 0.08% | 0.19% |
| | φ ↑ | 99.60% | 99.20% | 99.70% | 99.41% | 99.34% | 98.96% | 99.82% | 99.56% | 99.82% | 99.63% |
| SUSY | C2W ↓ | 0.25% | 0.82% | 0.22% | 0.74% | 3.50% | 3.40% | 0% | 0.78% | 0% | 0.73% |
| | W2C ↓ | 0.24% | 0.78% | 0.21% | 0.73% | 1.34% | 1.14% | 0% | 0.79% | 0% | 0.76% |
| | φ ↑ | 99.51% | 98.40% | 99.58% | 98.53% | 95.16% | 95.46% | 100% | 98.43% | 100% | 98.51% |
| HIGGS | C2W ↓ | 0.00% | 2.52% | 0% | 2.64% | | | 0% | 1.92% | 0% | 1.92% |
| | W2C ↓ | 0.00% | 2.56% | 0% | 2.63% | OOM | | 0% | 1.93% | 0% | 1.92% |
| | φ ↑ | 100.00% | 94.92% | 100% | 94.73% | | | 100% | 96.14% | 100% | 96.17% |
| Optdigits | C2W ↓ | 0.33% | 0.56% | 0.17% | 0.28% | 0.22% | 0.56% | 0.61% | 0.45% | 0.45% | 0.61% |
| | W2C ↓ | 0.56% | 0.61% | 0.28% | 0.50% | 0.28% | 0.22% | 0.22% | 0.33% | 0.28% | 0.39% |
| | W2W ↓ | 0.06% | 0.11% | 0.06% | 0% | 0.17% | 0.11% | 0.06% | 0.11% | 0.06% | 0.06% |
| | φ ↑ | 99.05% | 98.72% | 99.50% | 99.22% | 99.33% | 99.11% | 99.11% | 99.11% | 99.22% | 98.94% |
| Pendigits | C2W ↓ | 0.26% | 0.83% | 0.14% | 0.17% | 0.17% | 0.09% | 0.29% | 0.26% | 0.26% | 0.23% |
| | W2C ↓ | 0.14% | 0.43% | 0.11% | 0.17% | 0.26% | 0.37% | 0.17% | 0.20% | 0.23% | 0.20% |
| | W2W ↓ | 0.06% | 0.20% | 0.06% | 0.03% | 0.03% | 0.09% | 0.06% | 0.09% | 0.06% | 0.09% |
| | φ ↑ | 99.54% | 98.54% | 99.69% | 99.63% | 99.54% | 99.46% | 99.49% | 99.46% | 99.49% | 99.49% |
| Letter | C2W ↓ | 0.74% | 1.62% | 0.64% | 0.68% | 0.52% | 0.80% | 1.24% | 1.36% | 1.26% | 1.40% |
| | W2C ↓ | 0.82% | 0.88% | 0.78% | 0.80% | 0.58% | 0.62% | 1.06% | 1.42% | 1.06% | 1.38% |
| | W2W ↓ | 0.28% | 0.44% | 0.30% | 0.30% | 0.20% | 0.40% | 0.24% | 0.24% | 0.42% | 0.28% |
| | φ ↑ | 98.16% | 97.06% | 98.28% | 98.22% | 98.70% | 98.18% | 97.26% | 96.98% | 97.26% | 96.94% |
| Covtype | C2W ↓ | 0.98% | 2.37% | 1.78% | 1.78% | 0.11% | 0.61% | 1.94% | 2.04% | 1.94% | 1.96% |
| | W2C ↓ | 1.15% | 2.10% | 1.77% | 1.77% | 0.14% | 0.70% | 1.80% | 1.76% | 1.80% | 1.71% |
| | W2W ↓ | 0.04% | 0.09% | 0.07% | 0.07% | 0.02% | 0.03% | 0.06% | 0.07% | 0.06% | 0.07% |
| | φ ↑ | 97.83% | 95.44% | 96.38% | 96.38% | 99.74% | 98.66% | 96.19% | 96.13% | 96.20% | 96.26% |

**Definition 3** (Functional Similarity) *Given an input space $\mathcal{X}$, a model $T$, a model $\hat{T}$ online learned from $T$, and a dataset $D = \{y_i, \mathbf{a}_i\} \in \mathcal{X}$, the functional similarity $\phi$ between model $T$ and $\hat{T}$ is: $\phi = 1 - (r_{w2w} + r_{w2c} + r_{c2w})$ ,where $\phi$ is the leakage of learning.*

Due to the size limitations of the table, we have omitted OnlineGB from this table because its learning duration is excessively long, making it relatively meaningless compared to other methods. We compared iGBDT in adding 1 and 0.1% data instances, and DeltaBoost and MUinGBDT in deleting data. As shown in Table 7, we have a comparable model functionality in adding/deleting both 1 and 0.1%. In most cases, our online learned model reaches 98% similarity in both incremental learning and decremental learning.

## G BACKDOOR ATTACKING

**Experimental Setup.** In this evaluation, we randomly select a subset of the training dataset, and set first a few features to a specific value (trigger, e.g. 0 or greatest feature value) on these data instances, and then set the label to a target label (e.g., 0). In the testing dataset, we set all labels to the target label to compose a backdoor test dataset. In this setting, if the model has correctly learned the trigger and target label, it should achieve a high accuracy on backdoor test dataset.

## H MEMBERSHIP INFERENCE ATTACK

**Experimental Setup.** For over-fitting the model, we set the number of iteration $M$ to be 5, and split each dataset into three subsets: base dataset $D_{\text{base}}$ (49.9%), online dataset $D'$ (0.1%), and test dataset $D_{\text{test}}$ (50%). We first train a base model on $D_{\text{base}} + D'$. For this base model, the MIA should identify the data in $D'$ as part of the training dataset. Next, we perform decremental learning to delete $D'$ from the base model. After this process, the MIA should no longer identify the data in $D'$ as part of the training dataset. Finally, we add $D'$ back to the model using incremental learning. Following this, the MIA should once again identify the data in $D'$ as part of the training dataset.

**MIA Model.** By following the existing MIA methods Yan et al. (2023); Li et al. (2022); Carlini et al. (2022), we train an MIA model (binary classification) on the prediction probabilities of each class. Since the GBDT model is overfitted, the probability distributions of the training data should substantially differ from those of the unseen data (test data). Therefore, the MIA model can predict

whether a data sample is part of the training dataset based on its probability distribution. We sample 50% of $D_{\text{base}}$ and 50% of $D_{\text{test}}$ to train the MIA model. Then remaining 50% of $D_{\text{base}}$, the entire $D'$ and 50% of $D_{\text{test}}$ are used for evaluation.

# I EXTREMELY HIGH-DIMENSIONAL DATASETS

We include two dataset with more features / high dimensional: RCV1 and News20, which have 47,236 and 1,355,191 features respectively. For News20 dataset, the substantial high dimension causes segmentation fault on CatBoost and GPU out of memory (OOM) on thunderGBM. We omit the results from the other incremental/decremental method because infeasible running time and massive occupied memory. Table 9 shows the comparison of the training time and memory usage for our methods and other popular methods. Table 10 illustrates the incremental and decremental learning time of our method for two high dimensional dataset.

Table 8: Dataset specifications.

| Dataset | # Train | # Test | # Dim | # Class |
|---------|---------|--------|-------|---------|
| News20 | 5,000 | 14,996 | 1,355,191 | 2 |
| RCV1 | 20,242 | 677,399 | 47,236 | 2 |

Table 9: Comparison of the training time consumption and memory usage for RCV1 and News20.

| | Dataset | XGBoost | LightGBM | CatBoost | ThunderGMB (GPU) | Ours |
|---|---------|---------|----------|----------|------------------|------|
| Training Time (s) | RCV1 | 459.75 | 59.63 | 335.70 | 49.44 | 295.43 |
| | News20 | 637.02 | 28.42 | Seg. Fault | OOM | 225.73 |
| Memory ($MB$) | RCV1 | 3,008.28 | 2,922.32 | 263.63 | 1,913.05 | 185,851.72 |
| | News20 | 3,061.99 | 2,509.29 | Seg. Fault | OOM | 128,131.43 |

Table 10: The incremental/decremental learning time of the proposed method for RCV1 and News20. (ms, per tree, incre./decre.)

| Dataset | $|D'|$ | Learning Time (Ours) | Incremental Learning Speedup v.s. | | | | Learning Time (Ours) | Decremental Learning Speedup v.s. | | | |
|---------|--------|----------------------|---------|----------|----------|------------------|----------------------|---------|----------|----------|------------------|
| | | | XGBoost | LightGBM | CatBoost | ThunderGBM (GPU) | | XGBoost | LightGBM | CatBoost | ThunderGBM (GPU) |
| RCV1 | 1 | 21.431 | 214.5x | 27.8x | 156.6x | 23.1x | 19.268 | 238.6x | 30.9x | 174.2x | 25.7x |
| | 0.1% | 37.707 | 121.9x | 15.8x | 89.0x | 13.1x | 29.232 | 157.3x | 20.4x | 114.8x | 16.9x |
| | 0.5% | 39.428 | 116.6x | 15.1x | 85.1x | 12.5x | 48.218 | 95.3x | 12.4x | 69.6x | 10.3x |
| | 1% | 43.901 | 104.7x | 13.6x | 76.5x | 11.3x | 70.666 | 65.1x | 8.4x | 47.5x | 7.0x |
| News20 | 1 | 11.76 | 541.7x | 24.2x | - | - | 7.718 | 825.4x | 36.8x | - | - |
| | 0.1% | 17.113 | 372.2x | 16.6x | - | - | 12.363 | 515.3x | 23.0x | - | - |
| | 0.5% | 22.261 | 286.2x | 12.8x | - | - | 30.076 | 211.8x | 9.5x | - | - |
| | 1% | 23.469 | 271.4x | 12.1x | - | - | 37.825 | 168.4x | 7.5x | - | - |

Table 11: The approximation error of leave's score between the model after addition/delection and the model retrained from scratch. Appr. Error $= \frac{\sum_{\text{all trees}} \sum_{\text{all leaves}} \text{abs}(p_{\text{add/del}} - p_{\text{retrain}})}{\sum_{\text{all trees}} \sum_{\text{all leaves}} \text{abs}(p_{\text{retrain}})}$, where $p_{\text{add/del}}$ is the leave's score after adding/deleting, $p_{\text{retrain}}$ is the leave's score of the model retraining from scratch.

| | Adult | CreditInfo | SUSY | HIGGS | Optdigits | Pendigits | Letter | Covtype |
|---|-------|-----------|------|-------|-----------|-----------|--------|---------|
| Add 1 | 2.42% | 1.18% | 0.24% | 0.00% | 2.69% | 2.23% | 1.31% | 0.17% |
| Add 0.1% | 4.59% | 6.57% | 2.73% | 1.63% | 3.48% | 4.12% | 5.78% | 9.47% |
| Add 0.5% | 5.10% | 7.44% | 2.27% | 3.05% | 5.12% | 4.50% | 10.45% | 11.68% |
| Add 1% | 5.30% | 7.43% | 3.07% | 3.89% | 5.92% | 4.70% | 11.75% | 10.01% |
| Add 10% | 4.25% | 8.33% | 1.07% | 1.73% | 4.64% | 4.42% | 13.34% | 4.96% |
| Add 50% | 3.55% | 0.00% | 0.00% | 1.51% | 0.00% | 0.00% | 6.26% | 0.01% |
| Add 80% | 0.00% | 0.00% | 0.00% | 0.00% | 0.00% | 0.00% | 0.00% | 0.00% |
| Del 1 | 1.21% | 0.00% | 0.00% | 0.00% | 0.01% | 0.19% | 0.57% | 0.28% |
| Del 0.1% | 3.63% | 3.80% | 0.79% | 0.72% | 1.40% | 0.50% | 1.88% | 4.31% |
| Del 0.5% | 3.58% | 3.76% | 0.18% | 0.56% | 2.52% | 1.15% | 3.49% | 6.04% |
| Del 1% | 3.40% | 3.16% | 0.15% | 0.65% | 3.07% | 1.73% | 3.74% | 4.48% |
| Del 10% | 0.27% | 0.39% | 0.00% | 0.16% | 1.67% | 0.97% | 1.35% | 0.46% |
| Del 50% | 0.00% | 0.00% | 0.00% | 0.00% | 0.00% | 0.00% | 0.00% | 0.00% |
| Del 80% | 0.00% | 0.00% | 0.00% | 0.00% | 0.00% | 0.00% | 0.00% | 0.00% |

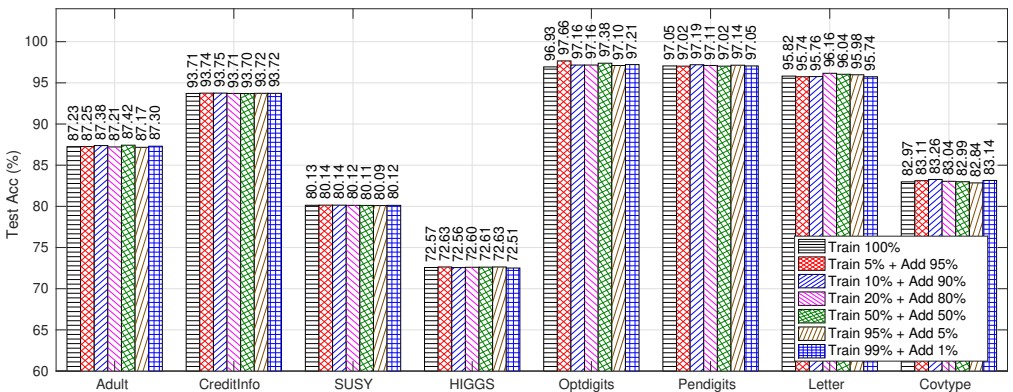

Figure 7: Different fine-tuning ratio.

## J APPROXIMATION ERROR OF LEAF SCORES

As mentioned in Section 3.2, outdated derivatives are used in gain computation to reduce the cost of updating derivatives. However, these outdated derivatives are only applied to nodes where the best split remains unchanged. When a sub-tree requires retraining, the derivatives are updated. Therefore, using outdated derivatives typically occurs when $|D'|$ is small, as fewer data modifications result in fewer changes to the best splits. Conversely, when more data is added or deleted, $|D'|$ becomes larger, increasing the likelihood of changes to the best splits in some nodes. As a result, the sub-trees are retrained, and the derivatives for the data reaching those nodes are updated.

To confirm the effect of using outdated derivatives during online learning, we report the result for the approximation error of leaf scores in Table 11. Appr. Error $= \frac{\sum_{\text{all trees}} \sum_{\text{all leaves}} \text{abs}(p_{\text{add/del}} - p_{\text{retrain}})}{\sum_{\text{all trees}} \sum_{\text{all leaves}} \text{abs}(p_{\text{retrain}})}$, where $p_{\text{add/del}}$ is the leaf score after adding/deleting, and $p_{\text{retrain}}$ is the leaf score of the model retraining from scratch. Please note that the retrained model has the same structure and split in all nodes of all trees as the model after adding/deleting, and we only update the latest residual and hessian to calculate the latest leaf score. When the number of added/deleted data increases, the error will increase because our method uses outdated derivatives if the best splits remain unchanged. When the number of add/delete is large enough, almost all nodes in the model will be retrained because their best splits have changed, so the error becomes 0.

## K ABLATION STUDY

In this section, we discuss the impact of different settings on the performance of our framework, e.g., time and accuracy.

**Size of Online Dataset $|D'|$.** Different sizes of online learning dataset $D'$ can have varying impacts on both the accuracy and time of the online learning process. Figure 7 shows the impact of different data addition settings on test accuracy. Across all datasets, our framework achieved nearly the same test accuracy, which validates the effectiveness of our online learning framework. Decremental learning also has similar results.

Figure 8 shows the influence of $|D_{in}|$ on incremental/decremental learning time. We only present the experiment on 2 datasets each for incremental/decremental learning, due to the results on other datasets show a similar trend. These results show that the online learning time

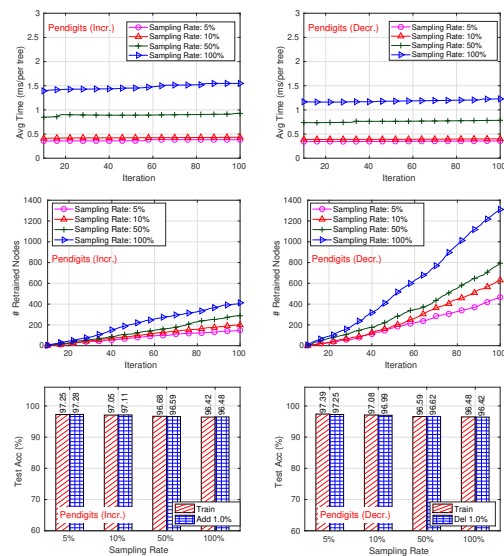

Figure 6: The impact of sampling rate on time, number of retrain nodes, and test accuracy during incremental/decremental learning.

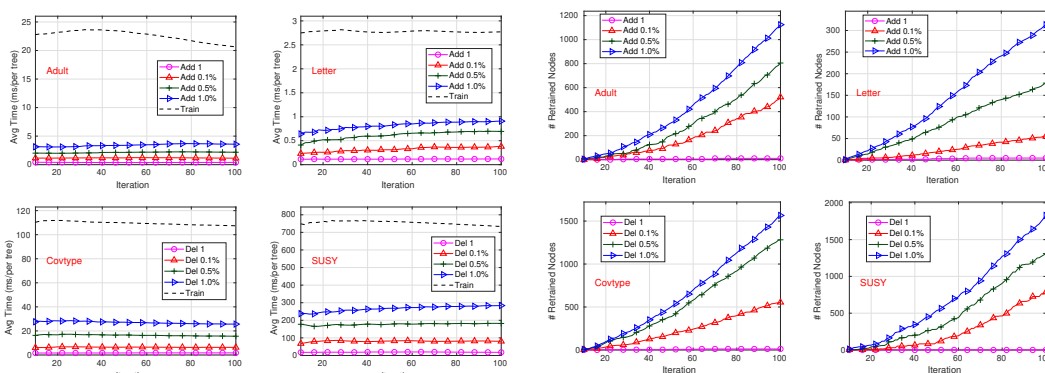

Figure 8: The impact of $|D'|$ on average learning time in incremental/decremental learning (top/bottom row).

Figure 9: The impact of $|D'|$ on the accumulated number of retrained nodes for each iteration in incr./decr. learning (top/bottom row).

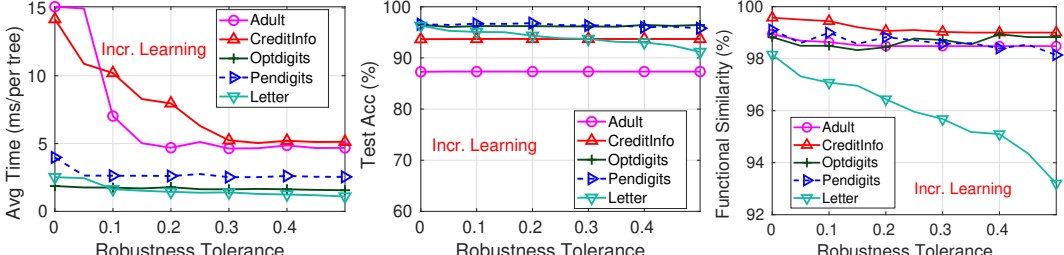

Figure 10: The impact of split robustness tolerance on the learning time, test accuracy, and model functional similarity $\phi$ in incremental learning.

increase when the size of $D_{in}$ increase. The reason is straightforward: as the size of $D_{in}$ increases, the model undergoes more significant changes, resulting in unstable splits. This leads to a greater number of sub-trees that require retraining, ultimately consuming more time. Figure 9 provides evidence to support this observation. It illustrates the accumulated number of retrained nodes – how many nodes need to be retrained. As the size of $D_{in}$ increases, the number of nodes that need to be retrained also increases. This leads to longer learning times.

**Split Random Sampling.** Split random sampling is designed to reduce the frequency of retraining by limiting the number of splits. As mentioned in Section 3.3, a smaller sampling rate leads to more stable splits, resulting in fewer nodes that require retraining and shorter online learning time. Figure 6 shows the impact of sampling rate $\alpha$ in split random sampling. The figures at the top demonstrate that when the sample rate is reduced, a smaller number of split candidates are taken into account, leading to an expected decrease in online learning time. However, there is no significant difference between 5% and 10% in the Pendigits dataset. The figures in the second row show the accumulated number of retrained nodes. It also shows that as the sample rate decreases, the splits become more stable, resulting in fewer nodes that require retraining. In Pendigits, since the number of nodes that require retraining is similar for 5% and 10%, it results in a minimal difference in the online learning time, as mentioned above. However, interestingly, for example in 100% sampling rate, although there are fewer retraining in incremental learning, it take more time during learning process, because incremental learning does not have derivatives of the data to be added. Therefore, more time is needed to calculate their derivatives. On the contrary, decremental learning can reuse the stored derivatives of the training process, resulting in less time. The bottom row shows the impact of the sampling rate on the test accuracy. The test accuracy remains almost identical across all sampling rates. Similar results can be observed in other datasets.

**Split Robustness Tolerance.** Split robustness tolerance aims to enhance the robustness of a split in online learning. As the observation in Figure 2, most best splits will be changed to second-best. Although the best split may change, we can avoid frequent retraining if we allow the split to vary

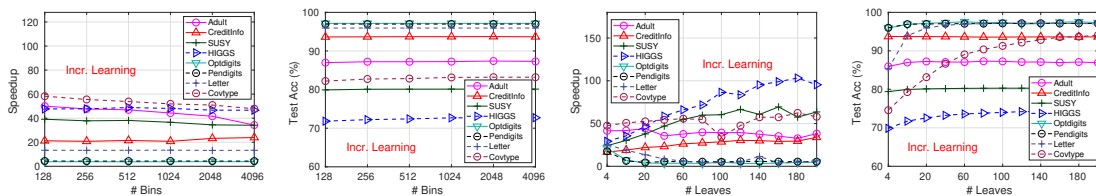

Figure 11: The impact of the # bins and # leaves on the acceleration factor of incremental learning (adding 1 data point).

within a certain range. For a node with $\lceil \alpha B \rceil$ potential splits, if the current split remains within the top $\lceil \sigma \alpha B \rceil$, we will continue using it. Here $\sigma$ ($0 \leq \sigma \leq 1$) is the robustness tolerance. Figure 10 illustrates the impact of split robustness tolerance $\sigma$ on learning time, test accuracy, and functional similarity $\phi$ in incremental learning. To obtain more pronounced experimental results, in this experiment, we set $|D'| = 1\% \times |D_{tr}|$. The figure on the left shows that the learning time decreases as the tolerance level increases. Although test accuracy changes only slightly (middle figure), the functional similarity $\phi$ drops significantly (right figure). For example, in the Letter dataset, $\phi$ drops about 5% from $\sigma = 0$ to $\sigma = 0.5$. This demonstrates that higher tolerance levels result in faster learning by avoiding retraining, but with a trade-off of decreased functional similarity. Therefore, we suggest $\sigma$ should not be greater than 0.15. Similar results can be obtained on decremental learning.

**Number of Bins and Leaves.** In online learning procedure, the number of bins and leaves also affects the online learning time. We report the impact of varying the number of bins $(128, 256, \cdots, 4096)$ and leaves $(4, 10, 20, 40, 60, \cdots, 200)$ on the acceleration factor of incremental learning (adding 1 data point) in Figure 11. The number of bins has few effect on both accuracy and the speed of online learning as shown in the top row of the figures. In terms of the number of leaves, when it exceeds 20, the accuracy tends to stabilize, except for Covtype, as shown in the bottom row of the figures. For smaller datasets (Adult, Optdigits, Pendigits, Letter), the more the number of leaves, the lower the acceleration factor for incremental learning. However, for larger datasets (CreditInfo, SUSY, HIGGS, Covtype), the more the number of leaves, the greater the acceleration is. Especially for HIGGS, the largest dataset in our experiments, the acceleration can be more than 100x.

