# OpenReview forum: "Online Gradient Boosting Decision Tree: In-Place Updates for Adding/Deleting Data"
_ICLR.cc/2025/Conference — Submitted to ICLR 2025_

### Official Review · Reviewer_remf · 2024-10-30

**Soundness:** 2
**Presentation:** 1
**Contribution:** 2
**Rating:** 3
**Confidence:** 3

**Summary:**

The authors propose an online learning framework that supports both incremental and decremental learning for Gradient Boosting Decision Trees (GBDT).

**Strengths:**

The authors have conducted an extensive set of experiments.

**Weaknesses:**

1. The motivation for the research is unclear. It seems to be solely because no one has studied it (line 75). If the motivation is closely related to the importance of GBDT, arguments such as *It outperforms deep learning models on many datasets in accuracy and provides interpretability for the trained models* lack supporting evidence.

2. The framework appears to be a minor improvement on existing research. Compared to traditional GBDT training methods, the difference in the new framework lies in utilizing the additivity of Formula 5 for node updates on Incremental & Decremental datasets (Section 2.3). The authors further discuss how to optimize time, which includes three parts. Section 3.1 also uses Incremental & Decremental datasets to update nodes rather than the entire dataset (this seems repetitive with the information conveyed in Section 2.3. If I'm wrong, please correct me.); Section 3.2 changes the frequency of executing Gradient Accumulation techniques, that is, update the derivatives only when retraining occurs instead of the traditional method that accumulates these gradients over multiple batches. Sections 3.3 and 3.4 reduce the time and resource consumption of online learning by changing from enumerating all potential splits to sampling a portion of the splits via introducing a sampling parameter $\alpha$. I acknowledge that these changes bring improvements to efficiency, but the inspiration provided by these minor innovations is limited.

3. The experimental section of the main text lacks discussion on important parameters and sampling. For example, there is a lack of discussion on the selection of the important parameter $\sigma$ and $\alpha$. The best split
changes in section 3.4 likely heavily depend on the distribution of the training data, as Figure 2 shows inconsistent changes in the best split points across different datasets. Therefore, whether $\sigma$ and $\alpha$  are data-dependent and how to determine them should be discussed in detail. And the main text lacks necessary conclusions of ablation studies. At the very least, the impact of the sampling method on accuracy should be pointed out. These have all been overlooked in main texts, and are replaced by a brief introduction in line 505.

4. Writing issues:

 - Citation issue in line 29.
 - Typo in Algorithm 3, line 4.
 - Incorrect citation format, such as line 29, 34, 39.
 - In line 533, it should be 'a novel'.
 - Experimental setup in section 3.4 needs more details, such as the repeated times and why 1% was chosen as the upper limit.

Overall, my main concern is that this work has very limited novelty and insight in terms of methodology (see my question 2). It appears to be an accumulation of minor modifications.I appreciate the authors' extensive experiments, but the structure of the experimental section needs significant revision. For example, while the experimental section covers various topics such as backdoor attacks (section 4.6), membership inference attack (section 4.7) and high-dimensional data (line 498), these experiments lack a progressive internal connection and convey repetitive information as section 4.3, 4.4 (i.e., the framework is effective in various scenarios). A suggestion is further insights about the stability and rationality of proposed method should be provided in the main text rather than in the appendix (see my question 3). Considering that the clarity of writing also needs improvement, I am inclined to recommend rejection.

**Questions:**

See my Weaknesses.

---

> ### Author Response · Authors · 2024-11-19
> **Official Comment by Authors (1/2)**
>
> Thank you very much for your valuable feedback. Here are our responses that could hopefully address your concerns.
>
> &nbsp;
>
> **W1:** Thank you for pointing it out. We appreciate the opportunity to clarify the motivation for our research. There is substantial evidence supporting the claim that GBDT can outperform deep learning models in certain tasks, particularly those involving tabular data. For example, studies [1, 2, 3, 4, 5] demonstrate scenarios where GBDT achieves superior accuracy due to its inherent ability to handle heterogeneity and sparsity in data more effectively than neural networks.
>
> Furthermore, GBDT models offer significant advantages in interpretability. Their structured, rule-based nature facilitates straightforward analysis, making them inherently more transparent compared to the often opaque nature of deep learning models. This transparency is further enhanced by the compatibility of GBDT with advanced interpretability tools such as SHAP (SHapley Additive exPlanations) [12] and LIME (Local Interpretable Model-agnostic Explanations) [7], which provide actionable insights into feature importance and model decision-making [6, 8, 9, 10, 11].
>
> We will expand the motivation section in our revised paper to include these points. We sincerely thank you for bringing this to our attention.
>
> **W2:**
> - Our method provides a unified in-place update mechanism for adding and deleting data in GBDT. To the best of our knowledge, this is the first method to support both incremental and decremental learning for GBDT. Conventional GBDT requires loading the entire dataset during the training process and does not permit adding or deleting data after training. Even existing online GBDT methods only support adding new data to the model but do not allow data deletion. This highlights the novelty of our method.
> - Section 2.3 provides an overview of our framework, while Section 3.1 delves into the detailed implementation of updating the model without touching the training data. However, due to page limitations, the detailed implementation is included in Appendix E. We will add a clarification to this section. Thank you for pointing it out.
> - We would like to argue that these optimizations are not minor innovations. For instance, we propose an adaptive lazy update inspired by gradient accumulation in deep learning. However, unlike gradient accumulation in deep learning, which introduces a new hyperparameter -- the number of accumulation steps [13, 14, 15] -- our adaptive lazy update does not introduce any hyperparameters. We confirm its effectiveness in Appendix K. Similarly, we present split candidate sampling and split robustness tolerance based on our observations as mentioned in the paper. We provide the motivation for the presented optimizations (Section 3, Appendices C and E), theoretical proof of the relationships among these optimizations (Definitions 1 and 2, Appendix D), performance gains achieved through these optimizations (Appendices F, I, J), and comprehensive experiments on their impact (Appendix K). Together, these demonstrate that our optimizations substantially improve efficiency while maintaining the model's performance.
>
>
> **W3:** We agree that the discussion on hyperparameters $\sigma$ and $\alpha$ is important. However, due to the extensive experiments included in our paper and the page limitations of the conference, we report the ablation study in Appendix K. This study covers (1) the size of the online dataset $|D'|$, (2) the split random sample rate $\alpha$, (3) the split robustness tolerance $\sigma$, (4) the number of bins $B$, and (5) the number of leaves $J$. We will add a discussion in the main content to highlight the significance of hyperparameters $\sigma$ and $\alpha$ and inform readers about the comprehensive experiments detailed in Appendix K.

---

> ### Author Response · Authors · 2024-11-19
> **Official Comment by Authors (2/2)**
>
> **W4:** We will address all these writing issues in the revised paper.
>
> &nbsp;
>
> Thank you for your feedback. We would like to emphasize that Section 4.3, 4.4, 4.6 and 4.7 are not repetitive:
> - Section 4.3 Test Error Rate: The goal of this experiment is to validate the accuracy of the trained model and the model after performing incremental and decremental learning.
> - Section 4.4 Batch Addition \& Removal: Unlike Section 4.3, which performs a one-time incremental/decremental learning, this experiment demonstrates that our method supports continual incremental/decremental learning. This capability is crucial for tasks involving datasets where the data or distribution may change over time.
> - Section 4.6 Verifying by Backdoor Attacking: In this experiment, unlike previous experiments where data samples are randomly selected, we verify that our method can add or remove specific data samples, such as poisoned (backdoor) data, which are not randomly chosen from the dataset.
> - Section 4.7 Verifying by Membership Inference Attack: This experiment provides a different perspective to confirm that data can be successfully deleted and added back.
>
> Our paper includes extensive experiments, and we will strive to make these sections clearer for readers. We will add this clarification in our revised paper and consider reorganizing the experimental sections if necessary.
>
> &nbsp;
>
> Thank you once again for your feedback. If you have any remaining concerns, please don't hesitate to let us know. We are more than happy to address and clarify them.
>
> &nbsp;
>
> [1] Grinsztajn, Léo, Edouard Oyallon, and Gaël Varoquaux. "Why do tree-based models still outperform deep learning on typical tabular data?." Advances in neural information processing systems 35 (2022): 507-520.
>
> [2] Shwartz-Ziv, Ravid, and Amitai Armon. "Tabular data: Deep learning is not all you need." Information Fusion 81 (2022): 84-90.
>
> [3] McElfresh, Duncan, et al. "When do neural nets outperform boosted trees on tabular data?." Advances in Neural Information Processing Systems 36 (2024).
>
> [4] Zoppi, Tommaso, Stefano Gazzini, and Andrea Ceccarelli. "Anomaly-based error and intrusion detection in tabular data: No DNN outperforms tree-based classifiers." Future Generation Computer Systems 160 (2024): 951-965.
>
> [5] Gorishniy, Yury, et al. "Revisiting deep learning models for tabular data." Advances in Neural Information Processing Systems 34 (2021): 18932-18943.
>
> [6] Štrumbelj, Erik, and Igor Kononenko. "Explaining prediction models and individual predictions with feature contributions." Knowledge and information systems 41 (2014): 647-665.
>
> [7] Ribeiro, Marco Tulio, Sameer Singh, and Carlos Guestrin. "" Why should i trust you?" Explaining the predictions of any classifier." Proceedings of the 22nd ACM SIGKDD international conference on knowledge discovery and data mining. 2016.
>
> [8] Lipovetsky, Stan, and Michael Conklin. "Analysis of regression in game theory approach." Applied stochastic models in business and industry 17.4 (2001): 319-330.
>
> [9] Konstantinov, Andrei V., and Lev V. Utkin. "Interpretable machine learning with an ensemble of gradient boosting machines." Knowledge-Based Systems 222 (2021): 106993.
>
> [10] Delgado-Panadero, Ángel, et al. "Implementing local-explainability in gradient boosting trees: feature contribution." Information Sciences 589 (2022): 199-212.
>
> [11] Blockeel, Hendrik, et al. "Decision trees: from efficient prediction to responsible AI." Frontiers in Artificial Intelligence 6 (2023): 1124553.
>
> [12] Lundberg, Scott M. and Su-In Lee. “A Unified Approach to Interpreting Model Predictions.” Neural Information Processing Systems (2017).
>
> [13] Aburass, Sanad, and Osama Dorgham. "Performance Evaluation of Swin Vision Transformer Model using Gradient Accumulation Optimization Technique." Proceedings of the Future Technologies Conference. Cham: Springer Nature Switzerland, 2023.
>
> [14] Hermans, Joeri R., Gerasimos Spanakis, and Rico Möckel. "Accumulated gradient normalization." Asian Conference on Machine Learning. PMLR, 2017.
>
> [15] Lin, Yujun, Song Han, Huizi Mao, Yu Wang and William J. Dally. "Deep Gradient Compression: Reducing the Communication Bandwidth for Distributed Training." International Conference on Learning Representations, ICLR, 2018.

---

> > ### Comment · Reviewer_remf · 2024-11-22
> >
> > Thank you for your response. After reading your reply, I acknowledge the effectiveness of the proposed framework. However, I maintain my assessment regarding the work's novelty. While I recognize that operations such as split candidate sampling are indeed new, the insights they provide (at least as presented in the main text) appear limited.
> > Furthermore, considering that the current manuscript's writing and experimental section organization require adjustment - such as many typos, wrong citation format, and the absence of crucial ablation experiments on key parameters - I maintain my score.
> >
> > However, I also acknowledge that I am not familiar with this specific field, which may have led to an inaccurate assessment of its innovativeness.

---

> ### Author Response · Authors · 2024-11-23
> **Reply to Reviewer remf (1/2)**
>
> Thank you for your reply. We sincerely appreciate your valuable feedback and want to express our gratitude for your dedication. We understand that reviewing a paper in an unfamiliar field requires substantial time and effort. We are happy to engage in further discussion with you and will do our best to address all your concerns.
>
> We would like to clarify that we will reorganized the ablation study on the key parameters you highlighted to the main text, while moving some of the other experiments to the appendix to better manage page space.
>
> For split candidate sampling, similar to the other sampling methods used on GBDT to enhance model performance, reduce overfitting, and improve computational efficiency [1, 2, 3], such as data sampling, feature sampling, one-side sampling, etc., our split candidate sampling is specifically designed to enhance computational efficiency while maintaining model performance. As shown in Figure 6 of the paper, varying the sampling rate yields a substantial speedup in online learning, with rates decreasing from 100% to 5% while maintaining identical performance. We also provide theoretical analyses in Definitions 1 and 2 to explain how split candidate sampling affects the robustness of the best split. To gain a deeper understanding of split candidate sampling, we conducted an experiment combining it with feature sampling, as both methods aim to reduce the number of splits. We report the impact of different feature sampling (10%, 20%, 50%, 100%) and split sampling (10%, 100%) on incremental learning (Add 1%) in Table R1. This table demonstrate that lower feature sampling rates result in faster training and incremental learning but with higher error rates. Conversely, lower split sampling rates achieve faster speeds with comparable error rates. Using 50% feature sampling and 10% split sampling strikes a good balance between speed and accuracy. All of these results shows that, our split candidate sampling in online learning is effective and efficient.
>
> We sincerely appreciate you effort and time on reviewing our work. If you have any specific concerns or have identified particular limitations, please do let us know. We are more than happy to conduct further experiments or analyses to address them, as we believe this will not only strengthen our paper but also provide more depth and value to our readers.
>
> Thank you once again for your time and for engaging in this discussion with us. Your input is invaluable, and we look forward to hearing your further suggestions and comments.
>
>
> [1] Chen, Tianqi, and Carlos Guestrin. "Xgboost: A scalable tree boosting system." Proceedings of the 22nd acm sigkdd international conference on knowledge discovery and data mining. 2016.
>
> [2] Ke, Guolin, et al. "Lightgbm: A highly efficient gradient boosting decision tree." Advances in neural information processing systems 30 (2017).
>
> [3] Han, Cuize, et al. "Scalable feature selection for (multitask) gradient boosted trees." International Conference on Artificial Intelligence and Statistics. PMLR, 2020.

---

> ### Author Response · Authors · 2024-11-23
> **Reply to Reviewer remf (2/2)**
>
> **Table R1**. The impact of different levels of feature sampling (10%, 20%, 50%, 100%) and split sampling (10%, 100%) on adding 1% of dataset.
>
> (Total Training Time (s) / Total Incremental Learning Time (s) / Error Rate)
>
> | Dataset|Split Sampling Rate $\alpha$|Feature Sampling Rate||| |
> |---|---|:---:|:---:|:---:|:---:|
> | ||10%|20%|50%|100% |
> | Adult|$\alpha = 1$|0.50 / 0.09 / 0.1324|0.70 / 0.07 / 0.1262|1.26 / 0.13 / 0.1262|2.05 / 0.21 / 0.1270 |
> | |$\alpha = 0.1$|0.47 / 0.08 / 0.1379|0.67 / 0.09 / 0.1296|1.18 / 0.07 / 0.1285|1.89 / 0.09 / 0.1285 |
> | ||||| |
> | CreditInfo|$\alpha = 1$|0.98 / 0.21 / 0.0646|1.14 / 0.16 / 0.0632|1.39 / 0.22 / 0.0628|1.76 / 0.34 / 0.0632 |
> | |$\alpha = 0.1$|0.89 / 0.21 / 0.0653|1.04 / 0.19 / 0.0628|1.29 / 0.15 / 0.0621|1.58 / 0.17 / 0.0627 |
> | ||||| |
> | SUSY|$\alpha = 1$|27.32 / 4.21 / 0.2293|31.58 / 4.56 / 0.2029|42.32 / 5.50 / 0.1987|59.40 / 7.35 / 0.1985 |
> | |$\alpha = 0.1$|28.17 / 4.17 / 0.2247|31.81 / 4.33 / 0.2035|42.88 / 5.23 / 0.1990|59.68 / 6.37 / 0.1989 |
> | ||||| |
> | HIGGS|$\alpha = 1$|63.96 / 11.81 / 0.3174|79.94 / 13.91 / 0.2949|116.40 / 14.89 / 0.2753|169.17 / 18.77 / 0.2723 |
> | |$\alpha = 0.1$|62.14 / 11.15 / 0.3160|79.78 / 13.45 / 0.2913|116.82 / 15.50 / 0.2765|170.39 / 18.20 / 0.2743 |
> | ||||| |
> | Optdigits|$\alpha = 1$|0.61 / 0.06 / 0.0329|0.72 / 0.15 / 0.0284|1.07 / 0.33 / 0.0284|1.65 / 0.58 / 0.0384 |
> | |$\alpha = 0.1$|0.59 / 0.05 / 0.0423|0.69 / 0.13 / 0.0301|0.99 / 0.29 / 0.0262|1.43 / 0.52 / 0.0278 |
> | ||||| |
> | Pendigits|$\alpha = 1$|0.91 / 0.06 / 0.1130|1.15 / 0.23 / 0.0380|1.68 / 0.59 / 0.0309|2.55 / 1.00 / 0.0346 |
> | |$\alpha = 0.1$|0.94 / 0.09 / 0.1147|1.12 / 0.19 / 0.0320|1.41 / 0.47 / 0.0275|1.95 / 0.75 / 0.0292 |
> | ||||| |
> | Letter|$\alpha = 1$|2.02 / 0.60 / 0.2356|2.20 / 0.30 / 0.0626|2.71 / 0.52 / 0.0416|3.45 / 0.92 / 0.0466 |
> | |$\alpha = 0.1$|1.65 / 0.50 / 0.3810|2.09 / 0.32 / 0.1354|2.61 / 0.53 / 0.0498|3.21 / 1.01 / 0.0422 |
> | ||||| |
> | Covtype|$\alpha = 1$|10.79 / 1.87 / 0.2436|13.98 / 1.80 / 0.2060|21.28 / 2.14 / 0.1809|35.39 / 3.52 / 0.1681 |
> | |$\alpha = 0.1$|9.98 / 1.78 / 0.2497|13.58 / 1.78 / 0.2155|21.10 / 1.75 / 0.1826|31.87 / 1.99 / 0.1706 |

---

> ### Author Response · Authors · 2024-11-26
> **Additional Motivation for Point W1**
>
> Thank you again for your valuable feedback on our paper. Inspired by the comment from `Reviewer PGvm`, we would like to provide additional motivation for our online GBDT to offer deeper insights into its importance. GBDT has been widely used in industry for applications, such as recommender systems and fraud detection. However, in dynamic and real-time applications where data distributions change frequently, traditional GBDT models require retraining from scratch whenever the dataset changes, which is computationally expensive and inefficient for real-time applications. Furthermore, although prior works have introduced either incremental or decremental learning methods (the baselines in our paper), they often neglect efficiency and do not handle both incremental and decremental learning simultaneously. To the best of our knowledge, our framework is the first to propose an efficient online GBDT supporting both incremental and decremental learning, making it ideal for scenarios where data distribution changes rapidly. Here are some practical examples that highlight the need for our framework:
>
> 1. **Real-Time Fraud Detection**: For Online payment platforms or banking systems, when new transactions occur, our method can incrementally learn patterns from the new data without retraining the model from scratch. Additionally, outdated data, such as older fraud patterns, may become irrelevant, and our method can remove them to improve accuracy and efficiency.
> 2. **Stock Market Prediction**: Financial markets generate new data continuously (e.g., prices, volumes, and news). Our method can incrementally learn these new trends in real time, keeping predictions up to date.
> 3. **Dynamic Recommender Systems**: User behaviors evolve constantly. When users produce new interactions, our model can incrementally learn these behaviors in real time, enabling personalized recommendations that reflect the most recent data.
> 4. **Data Deletion Compliance**: Decremental learning is particularly relevant for complying with regulations like GDPR (General Data Protection Regulation) and CCPA (California Consumer Privacy Act), which mandate the "right to be forgotten." When users request their data to be removed, our method can perform decremental learning to delete their data from the model in real time without retraining.
> 5. **Secure Federated Learning**: In federated GBDTs, participants may withdraw their data contributions. Our method allows this data to be removed in real time without retraining, ensuring both efficiency and privacy.
>
> These examples highlight the critical need for efficient online GBDTs in industrial applications handling with dynamic data. We will include these clarifications into our revised paper. Thank you again for bringing this to our attention. Please let us know if you have any further concerns, we would be more than happy to address them.

---

### Official Review · Reviewer_PGvm · 2024-11-01

**Soundness:** 4
**Presentation:** 4
**Contribution:** 4
**Rating:** 6
**Confidence:** 3

**Summary:**

This paper proposes an online learning framework for Gradient Boosting Decision Tree (GBDT) that supports both incremental and decremental learning. Instead of adding and removing trees, this paper considers an in-place update of decision trees to add or delete a small fraction of data. In detail, the proposed framework detects the changes in the best split of tree nodes and retrains the subtrees accordingly. The authors introduce a split candidate sampling and robust LogitBoost to avoid frequent retraining. Extensive experiments show that the proposed framework outperforms the existing methods regarding efficiency and effectiveness. Other experiments are also conducted, including verification of backdoor attacks, performance on extremely high-dimensional datasets, and ablation studies.

**Strengths:**

- The proposed framework, specifically the in-place update of decision trees, is novel and interesting. It represents a significant departure from traditional GBDT online learning, which often focuses on adding or removing entire trees. This can be computationally expensive and may lead to model instability. By contrast, this paper's in-place update mechanism offers a more granular and efficient approach by directly modifying existing tree structures. This granularity potentially allows for finer adaptation to data changes, leading to better performance with fewer resources.
- This paper demonstrates a high level of completeness in the experimental evaluation, not only including the running time, running memory, and test error but also the batch addition and deletion of data, data addition with more classes, verification using backdoor attacks, and so on. Experiments from various aspects show the effectiveness and efficiency of the proposed framework.
- The paper is well-written and easy to follow.

**Weaknesses:**

- While the proposed framework is undeniably novel and effective, its reliance on multiple heuristics and associated hyperparameters introduces complexity. Techniques like split candidate sampling and robust LogitBoost, while aimed at efficiency, require careful tuning, and should be studied case-by-case.
- No real-world datasets with varying data distributions are used in the experiments. The proposed framework's performance on such datasets would be interesting to see, as it would provide a more realistic evaluation of the framework's robustness and adaptability.
- The font sizes in Tables 1 and 2 are too small, making them difficult to read.

**Minor Issues:**

- Lines 59 and 199: "Eq. equation" -> "Eq." or "Equation"
- Line 77: Bagging samples instances with replacement, not generating disjoint subsets.
- Line 87: "major challenges of in-place online learning": these challenges are not specific to in-place online learning.
- Line 199: "with ascending depths" -> "layer by layer from the root to the leaves"
- Lines 271 and 272: "leading to a reduction in the frequency of retraining" -> "decreasing the frequency of retraining"
- Lines 273 and 274: "reduce the online learning time" -> "accelerate the online learning process"

**Questions:**

- Is the training in the experiment in Section 4.1 an offline training?
- How does the proposed framework perform under rapidly changing data distributions, where frequent retraining may be required to maintain performance, while lack of retraining could lead to poor performance?

---

> ### Author Response · Authors · 2024-11-19
>
> Thank you so much for your constructive feedback and support for our work. We deeply value your insightful comments. Here are our response to your questions:
>
> &nbsp;
>
> **W1:** We agree that the parameters of our method require careful tuning. To address this, we include a comprehensive ablation study in Appendix K. Additionally, we recommend settings such as $\alpha = 0.1$ and $\sigma = 0.1$, which provide substantial speedup for both incremental and decremental learning while maintaining the model's performance. Depending on the specific requirements of a task, $\alpha$ and $\sigma$ can be adjusted accordingly. Both parameters exhibit the same behavior: smaller values result in faster processing, but at the cost of reduced performance. We will include this discussion in the revised paper. Thank you for highlighting this point.
>
> **W2:** To confirm the performance of our methods on real-world datasets with varying data distributions, we conducted experiments on two time series datasets: (1) GlobalTemperatures [1]: This dataset records the average land temperatures from 1750 to 2015. (2) WebTraffic [2]: This dataset tracks hourly web requests to a single website over a span of five months.
>
> For this experiment, we constructed the input data $X$ using the time series values from the previous 15 time steps, with the goal of predicting the corresponding output value $y$. Initially, we randomly sample 10% of the data as the test dataset, with the remaining 90% used as the training dataset. Similar to Section 4.4, we evenly divided the training data into 10 subsets, each containing 10% of the training samples. It is important to note that we did not shuffle these time series datasets, meaning the 10 subsets were arranged sequentially from older to more recent data. We trained an initial model using the first subset, then incrementally added each subsequent subset one by one. After incorporating all training data, we sequentially removed each subset in reverse order. As expected, since the test dataset spans all time steps, the error rate decreases as more subsets are added to the model. This is because the model learns the updated distribution from the newly added subsets. After removing each subset, the error rate increases, reflecting the loss of information associated with the removed data and the model's adjustment to the remaining subsets. As shown in Table R1, these results confirm the effectiveness of our method in adapting to changing data distributions.
>
> **Table R1.** Error rate after every online learning step.
> |  | GlobalTemperatures ($\times 10^{-3}$) | WebTraffic ($\times 10^{-3}$) |
> |---|---|---|
> | Initial Train 10% | 4.1934 | 4.0984 |
> | Add 10% -> 20% | 2.5431 | 3.8383 |
> | Add 10% -> 30% | 2.1156 | 3.0296 |
> | Add 10% -> 40% | 2.0351 | 3.1297 |
> | Add 10% -> 50% | 1.9593 | 2.9149 |
> | Add 10% -> 60% | 1.8940 | 2.9525 |
> | Add 10% -> 70% | 1.8973 | 2.8682 |
> | Add 10% -> 80% | 1.8532 | 2.9024 |
> | Add 10% -> 90% | 1.8200 | 2.9141 |
> | Add 10% -> 100% | 1.7850 | 2.9049 |
> | Del 10% -> 90% | 1.8127 | 2.8432 |
> | Del 10% -> 80% | 1.9902 | 3.3453 |
> | Del 10% -> 70% | 2.0115 | 2.9007 |
> | Del 10% -> 60% | 2.1137 | 3.1288 |
> | Del 10% -> 50% | 2.0756 | 3.1187 |
> | Del 10% -> 40% | 2.1654 | 2.9539 |
> | Del 10% -> 30% | 2.1349 | 3.0132 |
> | Del 10% -> 20% | 2.4975 | 3.8429 |
> | Del 10% -> 10% | 3.6064 | 4.4339 |
>
> **W3:** Thank you for pointing it out. We will resize the tables to improve clarity and fix all minor issues in the revised paper.
>
> **Q1:** Yes, we first train a model offline, recording the time consumption and memory usage during the training process. Subsequently, we perform incremental learning and decremental learning, recording their respective time consumption.
>
> **Q2:** For the task under rapidly changing data distributions, we consider two scenarios:
> (1) Immediate Updates: The model requires updates after every incoming data sample to maintain optimal performance in real-time. In this scenario, our method is substantially faster than retraining the model from scratch.
> (2) Batch Updates: The model can wait and accumulate a batch of data samples before performing an update. In this scenario, our method supports batch updates and remains substantially faster than retraining the model from scratch, even when adding or removing 1% of the data samples.
> It is important to note that, unlike conventional online learning, the updates in this context can involve either adding or removing data.
>
> &nbsp;
>
> Thanks again for your valuable comments. We truly appreciate your detailed feedback.
>
> &nbsp;
>
> [1] GlobalTemperatures: https://www.kaggle.com/datasets/berkeleyearth/climate-change-earth-surface-temperature-data?resource=download&select=GlobalTemperatures.csv
>
> [2] WebTraffic: https://www.kaggle.com/datasets/raminhuseyn/web-traffic-time-series-dataset

---

> > ### Comment · Reviewer_PGvm · 2024-11-25
> >
> > I appreciate the authors' detailed responses to our questions and their presentation of additional experimental results on time series datasets, which effectively addressed my concerns.
> >
> > While the in-place updates for adding and deleting data in GBDT are both interesting and novel, Reviewer remf’s comments have prompted me to consider the broader motivation behind the proposed framework. In response to W1 raised by Reviewer remf, the authors emphasized the importance, advantages, and superiority of GBDT. However, this does not clarify the need for an algorithm capable of handling both data addition and deletion. The relevance of such a framework would be clearer in an online learning scenario where data is frequently and simultaneously added and removed, yet this case was not discussed. I recommend that the authors provide deeper insights into the motivation for their proposed framework.

---

> > > ### Author Response · Authors · 2024-11-26
> > >
> > > Thank you for raising this important question. We appreciate the opportunity to clarify the broader motivation for our proposed framework. GBDT has been widely used in industry for applications, such as recommender systems and fraud detection, due to its high accuracy and interpretability. However, in dynamic and real-time applications where data distributions change frequently, traditional GBDT models require retraining from scratch whenever the dataset changes, which is computationally expensive and inefficient for real-time applications. Furthermore, although prior works have introduced either incremental or decremental learning methods (the baselines in our paper), they often neglect efficiency and do not handle both incremental and decremental learning simultaneously. To the best of our knowledge, our framework is the first to propose an efficient online GBDT supporting both incremental and decremental learning, making it ideal for scenarios where data distribution changes rapidly. Here are some practical examples that highlight the need for our framework:
> > >
> > > 1. **Real-Time Fraud Detection**: For Online payment platforms or banking systems, when new transactions occur, our method can incrementally learn patterns from the new data without retraining the model from scratch. Additionally, outdated data, such as older fraud patterns, may become irrelevant, and our method can remove them to improve accuracy and efficiency.
> > > 2. **Stock Market Prediction**: Financial markets generate new data continuously (e.g., prices, volumes, and news). Our method can incrementally learn these new trends in real time, keeping predictions up to date.
> > > 3. **Dynamic Recommender Systems**: User behaviors evolve constantly. When users produce new interactions, our model can incrementally learn these behaviors in real time, enabling personalized recommendations that reflect the most recent data.
> > > 4. **Data Deletion Compliance**: Decremental learning is particularly relevant for complying with regulations like GDPR (General Data Protection Regulation) and CCPA (California Consumer Privacy Act), which mandate the "right to be forgotten." When users request their data to be removed, our method can perform decremental learning to delete their data from the model in real time without retraining.
> > > 5. **Secure Federated Learning**: In federated GBDTs, participants may withdraw their data contributions. Our method allows this data to be removed in real time without retraining, ensuring both efficiency and privacy.
> > >
> > > These examples highlight the critical need for efficient online GBDTs in industrial applications handling with dynamic data. Thank you again for bringing this important question to our attention. We will include these clarifications into our revised paper to provide deeper insights into our motivation. Additionally, we will share this enhanced motivation with `Reviewer remf` to ensure a comprehensive explanation. Please let us know if you have any further suggestions or concerns, we would be more than happy to address them.

---

### Official Review · Reviewer_Wpos · 2024-11-03

**Soundness:** 3
**Presentation:** 3
**Contribution:** 3
**Rating:** 8
**Confidence:** 4

**Summary:**

This paper aims to deal with the online learning task via optimized gradient boosting decision tree, which is an interesting research topic. And a novel online learning framework for GBDT supporting both incremental and decremental learning has been proposed. The whole paper is well organized with detailed formulation and theoretical analysis. Sufficient experiments have been given for model evaluation.

**Strengths:**

1. The research topic of optimizing the GBDT model is interesting, and the proposed method shows its novelty and performance.
2. The theoretical analysis of the proposed method in this paper is well expressed and proofed.

**Weaknesses:**

1. A detailed runtime complexity analysis of the proposed method is needed.
2. It is necessary to add an analysis on the impact of the number of base learners of the proposed model on the learning performance.

**Questions:**

1. There are also Online Boosting methods have been proposed, like OnlineBoost [1], OnlineAdaC2Classifier [1], and OnlineRUSBoost [1] have been proposed before, what is the difference between the proposed method and previous methods?

[1] B. Wang and J. Pineau, “Online Bagging and Boosting for Imbalanced Data Streams,” in IEEE Transactions on Knowledge and Data Engineering, vol. 28, no. 12, pp. 3353-3366.

2. The experiment results of the proposed method seem not the best on some datasets, please give a detailed explanation.
3. Although the parameter setting has been given, the parameter sensitivity analysis is still required.

---

> ### Author Response · Authors · 2024-11-19
> **Official Comment by Authors (1/4)**
>
> Thank you very much for your insightful comments and support of our work. We truly appreciate your valuable feedback. Here are our responses to your questions:
>
> &nbsp;
>
> **W1:** We compare the time complexity of retraining from scratch and our online learning approach in Table R1. Training a tree involves three key steps: Derivatives Computing, Gain Computing & Split Finding, and Prediction Computing. Let $B$ represent the number of bins, $J$ the number of leaves, $|D_{tr}|$ the number of training data points, and $|D'|$ the number of online learning data points ($|D'| \ll |D_{tr}|$).
> - **Derivatives Computing:** in retraining, each point is assigned to one of the $B$ bins, which take $O(\|D_{tr}\|)$ time. In our method, we optimize updates without touching training data, directly adding or subtracting derivatives for the online data points, which takes $O(\|D'\|)$ time.
> - **Gain Computing \& Split Finding:** in training, to identify the optimal split for each node, we compute the potential split gains for each bin. As a binary tree is constructed with $2J - 1$ nodes, the total computational complexity for split finding across the entire tree is $O(B(2J - 1)) = O(BJ)$. In our approach, Split Candidates Sampling reduces the number of split candidates from $B$ to $\alpha B$, where $\alpha$ denotes the split sample rate ($0 < \alpha \leq 1$). Additionally, let $P_\sigma$ represent the probability of a split change being within the robustness tolerance, indicating the likelihood that a node does not require retraining (with larger $\sigma$, $P_\sigma$ increases). If retraining is not required, the time complexity for checking a node is $O(|D'|)$. Conversely, if retraining is required, the complexity to retrain a node is $O(\alpha B)$. Consequently, the total time complexity for the entire tree is $O(J|D'| \cdot P_\sigma + J\alpha B \cdot (1-P_\sigma))$. For $P_\sigma \rightarrow 1$, no nodes require retraining, simplifying the complexity to $O(J|D'|)$. Conversely, for $P_\sigma \rightarrow 0$, all nodes require retraining, and the complexity becomes $O(J\alpha B)$.
> - **Predicted Value Computing:** during training, after the tree is built, the predicted value for each leaf is updated. This involves traversing the leaf for the data points that reach it, with the total number being equivalent to all training data points, resulting in a complexity of $O(|D_{tr}|)$. In our method, We update the predicted value only for leaves reached by at least one online data point, and adjust by adding or subtracting the impact of these online data points, resulting in a complexity of $O(|D'|)$.
>
> We present the time complexity comparison in Table R1 and will include this discussion in our revised paper. Thank you for bringing this to our attention.
>
> &nbsp;
>
>
> **Table R1.** Time complexity comparison between retraining and online learning.
> | Step | Training Time | Optimization | Online Learning Time |
> |---|---|---|---|
> | Derivatives Computing | $O(\|D_{tr}\|)$ | Update without Touching Training Data | $O(\|D'\|)$ |
> | Gain Computing \& Split Finding | $O(BJ)$ | Split Candidates Sampling, Split Robustness Tolerance | $O(J\|D'\|\cdot P_\sigma + J\alpha B\cdot(1-P_\sigma))$ |
> | Predicted Value Computing | $O(\|D_{tr}\|\log J)$ | Update without Touching Training Data | $O(\|D'\|)$ |

---

> ### Author Response · Authors · 2024-11-19
> **Official Comment by Authors (2/4)**
>
> **W2:** Thank you. We agree that the number of base learners is important in practical applications. We provide additional results for different numbers of base learners in Tables R2 and R3. Table R2 reports the test error rate after training, adding, and deleting base learners in GBDT models with varying iterations, demonstrating that our method achieves a comparable error rate across different iterations. Table R3 shows the time consumption for incremental and decremental learning, illustrating that our methods are substantially faster than retraining a model from scratch, particularly in cases where a single data sample is added or deleted.
>
> Additionally, to confirm that our method can effectively add and delete data samples across various iterations, we report results on backdoor attacks for different iterations, as shown in Table R4. These results confirm that our method successfully adds and removes data samples from the model across different numbers of iterations. We will include these experiments in the revised paper. Thank you for your helpful suggestions.
>
> **Table R2.** The test error rate after training, adding and deleting on GDBT models with various iterations.
> |||│|Adult||||│|CreditInfo||||│|Optdigits||||│|Pendigits||||│|Letter||||
> |---|---|:---:|:---:|---|---|---|:---:|:---:|---|---|---|:---:|:---:|---|---|---|:---:|:---:|---|---|---|:---:|:---:|---|---|---|
> |  |  | │ | 100 iterations | 200 iterations | 500 iterations | 1000 iterations | │ | 100 iterations | 200 iterations | 500 iterations | 1000 iterations | │ | 100 iterations | 200 iterations | 500 iterations | 1000 iterations | │ | 100 iterations | 200 iterations | 500 iterations | 1000 iterations | │ | 100 iterations | 200 iterations | 500 iterations | 1000 iterations |
> |Training|XGBoost|│|0.1270|0.1319|0.1379|0.1430|│|0.0630|0.0648|0.0663|0.0676|│|0.0418|0.0390|0.0412|0.0395|│|0.0397|0.0355|0.0352|0.0346|│|0.0524|0.0364|0.0356|0.0358|
> ||LightGBM|│|0.1277|0.1293|0.1260|0.1318|│|0.0635|0.0636|0.0644|0.0654|│|0.0334|0.0317|0.0334|0.0329|│|0.0355|0.0343|0.0340|0.0340|│|0.0374|0.0310|0.0296|0.0298|
> ||CatBoost|│|0.2928|0.2887|0.2854|0.2843|│|0.1772|0.1765|0.1765|0.1765|│|0.0618|0.0396|0.0293|0.0248|│|0.0440|0.0365|0.0281|0.0257|│|0.0655|0.0406|0.0252|0.0186|
> ||Ours|│|0.1276|0.1265|0.1294|0.1325|│|0.0629|0.0632|0.0639|0.0648|│|0.0307|0.0251|0.0239|0.0239|│|0.0294|0.0280|0.0277|0.0277|│|0.0418|0.0318|0.0256|0.0246|
> |─|─|┼|─|─|─|─|┼|─|─|─|─|┼|─|─|─|─|┼|─|─|─|─|┼|─|─|─|─|
> |Ours(Incr.Learning)|Add1|│|0.1275|0.1271|0.1287|0.1323|│|0.063|0.0635|0.0638|0.0644|│|0.0295|0.0262|0.0239|0.0239|│|0.0297|0.0275|0.0275|0.0275|│|0.0404|0.0330|0.0266|0.0260|
> ||Add0.1%|│|0.1269|0.1287|0.1313|0.1325|│|0.0626|0.0633|0.0631|0.0638|│|0.0295|0.0256|0.0256|0.0256|│|0.0297|0.0275|0.0277|0.0277|│|0.0406|0.0322|0.0250|0.0240|
> ||Add0.5%|│|0.1294|0.1276|0.1298|0.1316|│|0.0632|0.0629|0.0633|0.0648|│|0.029|0.0262|0.0256|0.0256|│|0.0295|0.0266|0.0283|0.0283|│|0.0394|0.0326|0.0270|0.0256|
> ||Add1%|│|0.1267|0.1279|0.1287|0.1337|│|0.0632|0.0630|0.0639|0.0646|│|0.0262|0.0228|0.0228|0.0228|│|0.0283|0.0272|0.0275|0.0277|│|0.044|0.0310|0.0246|0.0242|
> |─|─|┼|─|─|─|─|┼|─|─|─|─|┼|─|─|─|─|┼|─|─|─|─|┼|─|─|─|─|
> |Ours(Decr.Learning)|Del1|│|0.1276|0.1266|0.1294|0.1324|│|0.0628|0.0632|0.0640|0.0647|│|0.0306|0.0251|0.0239|0.0239|│|0.0295|0.0283|0.0280|0.0280|│|0.0416|0.0318|0.0260|0.0242|
> ||Del0.1%|│|0.1284|0.1273|0.1288|0.1321|│|0.0633|0.0634|0.0640|0.0648|│|0.0295|0.0256|0.0245|0.0245|│|0.0283|0.0280|0.0280|0.0280|│|0.0432|0.0336|0.0272|0.0246|
> ||Del0.5%|│|0.1295|0.1266|0.1280|0.1327|│|0.0634|0.0631|0.0644|0.0646|│|0.0301|0.0245|0.0239|0.0239|│|0.0303|0.0289|0.0283|0.0283|│|0.0432|0.0320|0.0258|0.0244|
> ||Del1%|│|0.1295|0.1281|0.1290|0.1313|│|0.0632|0.0633|0.0638|0.0654|│|0.0273|0.0239|0.0234|0.0234|│|0.0303|0.0292|0.0280|0.0280|│|0.0424|0.0328|0.0270|0.0252|

---

> ### Author Response · Authors · 2024-11-19
> **Official Comment by Authors (3/4)**
>
> **Table R3.** The Total training, incremental or decremental learning time (in seconds).
> |  | | │ | Adult |  |  |  | │ | CreditInfo |  |  |  | │ | Optdigits |  |  |  | │ | Pendigits |  |  |  | │ | Letter |  |  |  |
> |---|---|:---:|:---:|---|---|---|:---:|:---:|---|---|---|:---:|:---:|---|---|---|:---:|:---:|---|---|---|:---:|:---:|---|---|---|
> |  |  | │ | 100 iterations | 200 iterations | 500 iterations | 1000 iterations | │ | 100 iterations | 200 iterations | 500 iterations | 1000 iterations | │ | 100 iterations | 200 iterations | 500 iterations | 1000 iterations | │ | 100 iterations | 200 iterations | 500 iterations | 1000 iterations | │ | 100 iterations | 200 iterations | 500 iterations | 1000 iterations |
> | Training | XGBoost | │ | 9.467 | 19.128 | 43.064 | 103.767 | │ | 13.314 | 34.619 | 77.706 | 78.845 | │ | 0.752 | 1.385 | 2.598 | 5.271 | │ | 0.574 | 1.743 | 3.225 | 5.976 | │ | 1.171 | 3.647 | 8.097 | 14.597 |
> |  | LightGBM | │ | 0.516 | 0.926 | 1.859 | 3.775 | │ | 1.836 | 2.081 | 4.737 | 8.504 | │ | 0.106 | 0.164 | 0.248 | 0.462 | │ | 0.131 | 0.196 | 0.351 | 0.516 | │ | 0.203 | 0.376 | 0.758 | 1.342 |
> |  | CatBoost | │ | 1.532 | 2.646 | 5.805 | 10.974 | │ | 3.447 | 5.467 | 12.002 | 13.339 | │ | 0.177 | 0.458 | 1.160 | 2.360 | │ | 0.183 | 0.399 | 1.104 | 1.986 | │ | 0.232 | 0.524 | 1.475 | 3.196 |
> |  | Ours | │ | 2.673 | 3.289 | 7.466 | 14.509 | │ | 1.818 | 3.005 | 5.391 | 14.122 | │ | 0.276 | 0.573 | 1.444 | 2.874 | │ | 0.368 | 0.592 | 1.978 | 3.990 | │ | 0.352 | 0.357 | 1.284 | 1.798 |
> | ─ | ─ | ┼ | ─ | ─ | ─ | ─ | ┼ | ─ | ─ | ─ | ─ | ┼ | ─ | ─ | ─ | ─ | ┼ | ─ | ─ | ─ | ─ | ┼ | ─ | ─ | ─ | ─ |
> | Ours(Incr. Learning) | Add 1 | │ | 0.035 | 0.071 | 0.167 | 0.328 | │ | 0.114 | 0.125 | 0.244 | 0.616 | │ | 0.011 | 0.031 | 0.118 | 0.285 | │ | 0.014 | 0.045 | 0.142 | 0.227 | │ | 0.016 | 0.018 | 0.206 | 0.464 |
> |  | Add 0.1% | │ | 0.105 | 0.167 | 0.402 | 0.859 | │ | 0.249 | 0.307 | 0.661 | 2.402 | │ | 0.015 | 0.031 | 0.106 | 0.311 | │ | 0.026 | 0.059 | 0.187 | 0.347 | │ | 0.040 | 0.070 | 0.483 | 0.807 |
> |  | Add 0.5% | │ | 0.212 | 0.383 | 0.937 | 2.463 | │ | 0.321 | 0.593 | 1.502 | 4.670 | │ | 0.029 | 0.039 | 0.137 | 0.335 | │ | 0.042 | 0.062 | 0.194 | 0.411 | │ | 0.067 | 0.127 | 0.537 | 0.979 |
> |  | Add 1% | │ | 0.344 | 0.670 | 1.747 | 3.904 | │ | 0.383 | 0.789 | 2.255 | 6.369 | │ | 0.043 | 0.042 | 0.146 | 0.344 | │ | 0.053 | 0.067 | 0.202 | 0.435 | │ | 0.128 | 0.176 | 0.657 | 1.207 |
> | ─ | ─ | ┼ | ─ | ─ | ─ | ─ | ┼ | ─ | ─ | ─ | ─ | ┼ | ─ | ─ | ─ | ─ | ┼ | ─ | ─ | ─ | ─ | ┼ | ─ | ─ | ─ | ─ |
> | Ours(Decr. Learning) | Del 1 | │ | 0.034 | 0.128 | 0.177 | 0.179 | │ | 0.055 | 0.265 | 0.359 | 0.342 | │ | 0.010 | 0.007 | 0.037 | 0.092 | │ | 0.015 | 0.012 | 0.067 | 0.165 | │ | 0.014 | 0.007 | 0.007 | 0.011 |
> |  | Del 0.1% | │ | 0.103 | 0.305 | 0.541 | 0.549 | │ | 0.153 | 0.595 | 0.729 | 0.665 | │ | 0.014 | 0.011 | 0.045 | 0.115 | │ | 0.025 | 0.020 | 0.089 | 0.185 | │ | 0.058 | 0.017 | 0.021 | 0.021 |
> |  | Del 0.5% | │ | 0.222 | 0.753 | 1.481 | 1.467 | │ | 0.251 | 0.941 | 1.217 | 1.220 | │ | 0.029 | 0.024 | 0.065 | 0.123 | │ | 0.041 | 0.038 | 0.106 | 0.198 | │ | 0.103 | 0.035 | 0.041 | 0.038 |
> |  | Del 1% | │ | 0.379 | 1.297 | 2.033 | 2.464 | │ | 0.355 | 1.375 | 2.556 | 2.694 | │ | 0.046 | 0.035 | 0.075 | 0.132 | │ | 0.057 | 0.050 | 0.119 | 0.209 | │ | 0.134 | 0.051 | 0.060 | 0.056 |
>
> &nbsp;
>
> **Table R4.** Accuracy for clean test dataset and attack successful rate for backdoor test dataset. (Section 4.6)
> | Iteration | Dataset | │ | Train Clean |  | │ | Train Backdoor |  | │ | Add Backdoor |  | │ | Remove Backdoor |  |
> |---|---|:---:|---|---|:---:|---|---|:---:|---|---|:---:|---|---|
> |  |  | │ | Clean | Backdoor | │ | Clean | Backdoor | │ | Clean | Backdoor | │ | Clean | Backdoor |
> | 200 | Optdigits | │ | 97.49% | 8.85% | │ | 97.55% | 100.00% | │ | 97.27% | 100.00% | │ | 97.49% | 8.80% |
> |  | Pendigits | │ | 97.28% | 5.06% | │ | 97.25% | 100.00% | │ | 97.25% | 100.00% | │ | 100.00% | 11.67% |
> |  | Letter | │ | 96.82% | 2.90% | │ | 96.64% | 100.00% | │ | 96.56% | 100.00% | │ | 96.74% | 2.56% |
> | ─ | ─ | ┼ | ─ | ─ | ┼ | ─ | ─ | ┼ | ─ | ─ | ┼ | ─ | ─ |
> | 500 | Optdigits | │ | 97.61% | 8.63% | │ | 97.49% | 100.00% | │ | 97.72% | 100.00% | │ | 97.66% | 8.57% |
> |  | Pendigits | │ | 97.23% | 5.06% | │ | 97.14% | 100.00% | │ | 97.28% | 100.00% | │ | 97.25% | 5.63% |
> |  | Letter | │ | 97.44% | 5.18% | │ | 97.36% | 100.00% | │ | 97.14% | 100.00% | │ | 97.14% | 3.56% |
> | ─ | ─ | ┼ | ─ | ─ | ┼ | ─ | ─ | ┼ | ─ | ─ | ┼ | ─ | ─ |
> | 1000 | Optdigits | │ | 97.61% | 8.63% | │ | 97.77% | 100.00% | │ | 97.72% | 100.00% | │ | 97.83% | 10.30% |
> |  | Pendigits | │ | 97.23% | 5.00% | │ | 97.11% | 100.00% | │ | 97.28% | 100.00% | │ | 97.25% | 4.46% |
> |  | Letter | │ | 97.66% | 5.18% | │ | 97.38% | 100.00% | │ | 97.52% | 100.00% | │ | 97.42% | 11.18% |

---

> ### Author Response · Authors · 2024-11-19
> **Official Comment by Authors (4/4)**
>
> **Q1:** Thank you for your insightful question. We outline the key differences between our method and the previous method [1] as follows:
>
> 1. Different Target: To the best of our knowledge, our method is the first to consider in-place updates for both adding and removing data. The previous method [1] is similar to other online methods [2, 3], focusing solely on adding new data to the model without supporting data deletion. However, the distribution of datasets in certain tasks can change over time. Simply adding new data to adapt to a new distribution is often insufficient; it may also require the removal of outdated data to maintain accuracy and relevance.
>
> 2. Different Setting: Our method emphasizes updating models dynamically by adding or removing data, whereas method [1] addresses data imbalance by extending batch-mode algorithms like bagging and boosting to their online cost-sensitive versions, handling class imbalances in streaming data.
>
> 3. Complexity and Practical Implications: Our method incorporates several optimizations to reduce computational costs while maintaining high accuracy, making it scalable for large datasets. In contrast, method [1] may face scalability challenges due to the repetitive resampling and retraining of models with updated weight distributions.
>
> In summary, our method is more scalable and efficient, enabling seamless updates to trained models through incremental and decremental learning for both adding and removing data. We will include this discussion in the revised paper.
>
> **Q2:** Thank you for your question. As shown in Table 4 (in paper), after initial training across 10 datasets, our method achieves the best error rate on two datasets (CreditInfo and Pendigits) and obtains the second-best error rate on five datasets (Adult, HIGGS, Optdigits, Letter, and Abalone). Additionally, our experiments reveal that no single method achieves the best error rate across all datasets. To provide a comparative measure, we calculate the mean absolute error (MAE) relative to the best error rate for each method: $\text{MAE}\_{\text{method}} = \frac{\sum_\text{datasets} \text{Error Rate}_\text{dataset} - \text{Best Error on Dataset}}{\text{Number of Datasets}}$ as presented in Table R5. Our Method achieves the lowest MAE of 0.0028, demonstrating our method's superior performance and robustness relative to the other methods across the tested datasets.
>
> **Table R5.** Mean absolute error (MAE) relative to the best error rate for each method:  $\text{MAE}\_{\text{method}} = \frac{\sum_\text{datasets} \text{Error Rate}_\text{dataset} - \text{Best Error on Dataset}}{\text{Number of Datasets}}$.
> | Methods | MAE |
> |---|---|
> | XGBoost | 0.0093 |
> | LightGBM | 0.0030 |
> | CatBoost | 0.1070 |
> | ThunderGMB (GPU) | 0.1821 |
> | Ours | **0.0028** |
>
> **Q3:** We agree that parameter sensitivity analysis is crucial for understanding performance under different parameter settings. We have included a comprehensive ablation study on various settings in Appendix K.
> - Split Random Sampling: Split random sampling is designed to reduce the frequency of retraining by limiting the number of splits. A smaller sampling rate, $\alpha$, results in more stable splits, leading to fewer nodes requiring retraining and shorter online learning times. While accuracy remains largely unaffected, a substantial speedup is observed for $\alpha = 5%$ and $10%$ across datasets. Therefore, we recommend setting $\alpha = 10%$.
> - Split Robustness Tolerance: Split robustness tolerance enhances the robustness of splits during online learning. Higher tolerance levels result in faster learning by reducing the need for retraining but come with a trade-off of decreased functional similarity. Based on our findings, we suggest that $\sigma$ should not exceed 0.15.
> - Number of Bins: The number of bins has minimal impact on both accuracy and the speed of online learning.
> - Number of Leaves: When the number of leaves exceeds 20, accuracy tends to stabilize. Increasing the number of leaves further results in greater acceleration without significant loss of accuracy.
> - Size of Online Dataset $|D'|$: Online learning time increases as the size of $|D'|$ grows.
>
> We will provide clearer clarifications and explanations regarding our ablation study in the revised paper. Thank you for highlighting this point.
>
> &nbsp;
>
> Thank you again for your detailed feedback. We greatly appreciate your valuable insights.
>
> &nbsp;
>
> [1] B. Wang and J. Pineau, “Online Bagging and Boosting for Imbalanced Data Streams,” in IEEE Transactions on Knowledge and Data Engineering, vol. 28, no. 12, pp. 3353-3366.
>
> [2] Leistner, Christian, et al. "On robustness of on-line boosting-a competitive study." 2009 IEEE 12th International Conference on Computer Vision Workshops, ICCV Workshops. IEEE, 2009.
>
> [3] Zhang, Chongsheng, et al. "On incremental learning for gradient boosting decision trees." Neural Processing Letters 50 (2019): 957-987.

---

> > ### Comment · Reviewer_Wpos · 2024-11-25
> >
> > I have read the response and revised paper carefully, the author has addressed the issues I concerned, I have no further suggestion, and I will keep the current score.

---

> > > ### Author Response · Authors · 2024-11-26
> > >
> > > Thank you for your insightful comments and support. We will include all the new results and discussions in our revised paper. Your feedback has been invaluable in improving the quality of our work. Please do not hesitate to let us know if you have any concerns or questions.

---

### Meta-Review · Area_Chair_PK7Z · 2024-12-19

**Metareview:**

This paper proposes an online learning framework for Gradient Boosting Decision Tree (GBDT). This framework supports both incremental and decremental learning. An in-place update of decision trees to add or delete a small fraction of data is notably proposed. Some optimisation schemes were proposed with a theoretical analysis on the relationship with the hyperparameters. Some experiments were proposed to evaluate the behavior of the method, in particular we can notice the evaluation of of backdoor attacks, performance on extremely high-dimensional datasets, and ablation studies.


Strengths:
- optimizing Gradient Boosting Decisions trees is interesting,
- novelty of in-place update of DT,
- solid theoretical contribution,
- completeness of the empirical evaluation,
- paper easy to follow.

Weaknesses:
- the method relies on multiple heuristics and hyper-parameters,
- improvement is marginal with respect of state of the art,
- lack of run time analysis,
- lack of analysis of weak learners,
- no-real datasets,
- motivation insufficient,
- adjustments are needed.


Authors did a strong effort to address reviewers concerned, in particular many additional experiments were given.
During the discussion, the novelty of the approach was discussed, a conclusion is that the idea of in-place update instead of adding or removing entire trees seems natural but the tradeoff between computational cost and performance has not been studied before which is interesting.
However, the motivation was still an issue on the need for both adding and removing data in GBDT is not convincing. Authors authors did not provide any experimental results for the scenario where data is added and removed at the same time. Instead, the experiments are done separately for adding and removing data, which is not convincing.
The experimental evaluation could be better structured.

These points have made the paper to be evaluated below the bar. I propose then rejection.

**Additional Comments On Reviewer Discussion:**

Reviewers have acknowledged the answers and efforts made by the authors.
Reviewer remf, who was among the most negatives, maintained his reservations on the novelty of the work and the presentation, he found a lack of rigor, and was not convinced by the motivation of simultaneous adding and removing data in GBDT.
Reviewer PGvm agrees on the reservations on the motivation of the work for the same reason as above. He also thinks that the paper could be reorganized.
There was discussion on the novelty of the contribution, remf was that convinced, PGvm was more positive but agreed that the principle of addition/removing was natural but that the authors did an original work in performing of a study on this.

Overall, the motivation was still an important issue which convinced me to propose rejection.

---

### Decision · Program_Chairs · 2025-01-22

Reject